# The representation of omitted sounds in the mouse auditory cortex

Janek Peters[1,2], Zhongnan Cai[1,2], Maxime van Veghel[1], Andreas Knoben [1], Maikel Simon[1], Samuel Arends[1], Francesco Paolo Battaglia [1] & Bernhard Englitz [1] ✉

Humans and animals use predictions to optimize their behavior, however, the underlying neuronal implementation remains elusive. We address this using omitted sounds on the macro- and microscale of the auditory cortex of female, normal hearing mice using high-speed imaging. Neuronal responses to the omission of expected sounds were time-locked to expected stimulus onset, localized to layer 1-4 of higher auditory area (Temporal Association Area, TeA), and continued to rise until the following stimulus. The omission responses differed from offset and deviant responses in their temporal shape, size and spatial localization. Omissions and sequence statistics correlated with behavioral changes by timed pupil dilation and rapid facial motions. While stimulus responses showed partial entrainment, omission responses maintained a distinct, unentrained shape. The localized omission response in TeA is consistent with a hierarchical organization of predictive processing. However, the continued rise suggests an integrated, absolute prediction error, instead of a direct representation of prediction or prediction error, which would terminate with the omission.

The sensory world is full of predictable relationships in time and space. Previous research has convincingly demonstrated that humans and animals exploit this predictability to optimize their behavior, i.e., in generating faster and more accurate responses in complex decision tasks[1]. It has been suggested that predictable information and its use are so ubiquitous that this may be one of the central principles of brain organization[2,3]. The underlying neural representation forms an entire field of study in neuroscience that has remained, however, without a clear consensus[4–7].

In humans, a well-studied correlate of prediction-related processes is the mismatch negativity (MMN[8]), which can be reliably detected in EEG (electroencephalography) and MEG (magnetoencephalography) recordings and whose origin can be localized partly to the auditory cortex[9–11]. The MMN is reliably elicited when there is an unexpected change in a predictable sequence of sounds. Tracing the origin of the MMN in animal studies has provided a plethora of insights and has relied largely on deviants in predictable sequences[12–17]. In multiple species, neuronal responses to such deviant stimuli, in particular an unexpected change in pure tone frequency, have been shown to be elevated and organized hierarchically, with more prediction-related responses found in higher levels of the neuraxis[14,18,19]. Relating these results to the corresponding theories of predictive processing has been difficult, in part because these theories distinguish between predictions and prediction errors, which are hard to disentangle in classical deviant stimulus paradigms.

A promising alternative is to instead unexpectedly omit a predictable stimulus. It has been argued that this provides a clearer perspective on the putative top-down, predictive processes, as no bottom-up stimulus information arrives from the periphery[20–23]. Recent studies from our lab and others have demonstrated the existence of well-timed, positive neural responses to omitted sounds[12,24–26], while another study demonstrated such responses only in LFPs in the AC[27]. Responses to end-of-sequence omissions in very slow sequences (0.5 Hz) have been described as *echo responses* for slow, repeating

[1]Computational Neuroscience Lab, Donders Institute for Brain, Cognition and Behavior, Radboud University, Nijmegen, The Netherlands. [2]These authors contributed equally: Janek Peters, Zhongnan Cai. ✉e-mail: bernhard.englitz@donders.ru.nl

sequences[28,29], however, without clear spatial localization in the auditory cortex. We recently demonstrated a substantial increase in omission responses in the awake state compared with anesthesia[25]. This renders omission responses a possible correlate of predictive processes of the neural representation.

Previous studies used electrophysiology, whose temporal resolution could resolve the fast stimulus sequences in the typically used paradigms[16,30–32]. However, electrophysiology provides only limited insight into spatial arrangement on the macro- and microlevel. Imaging methods can provide complete spatial coverage and superior spatial resolution; however, have previously been limited in speed and signal-to-noise when using the GCaMP6-family of calcium indicators[33,34]. Here, we leveraged the substantial improvements of more recent Ca²⁺-indicators, using jGCaMP8m, which provides temporal fidelity up to 100 Hz (~2.5 ms rise time and 50% decay time ~40 ms see Extended Data Table 6[35]) and manifold improved brightness. This enables the study of spatial organization of omission responses from the whole auditory cortex to the cellular level, including the dependence on cortical depth for fast acoustic sequences using aligned 2-photon and widefield imaging[36].

We identify a specific subarea of the auditory cortex of mice, the posterior and medial part of the temporal association area (TeA), that dominantly shows responses to omitted tones, while sound responses are much smaller than in the primary areas, where the converse is observed. Omission responses are most prominent in L2-4 and decrease below. TeA has recently been noted for its late responses and top-down-like responses[37]. The omission responses show a time-locked increase at the beginning of the omitted sound, which continues to increase until the next stimulus. If a second omission follows the first omission, an additional, time-locked increase in response

occurs. The omission-responsive region does not coincide with areas that show substantial repetition suppression or prediction error in traditional oddball sequences. The stimulus response increases in size upon repeated stimulus presentations (*entrainment*) in the omission-responsive region. The omission response also increases, however, much more slowly, and maintains a very distinct shape from the stimulus response, and can thus not be explained by entrainment. Omissions elicit a timed pupil dilation during passive listening, indicating that the animal noticed the change in stimulus sequence. The strength of rapid facial motions adapts to repeated acoustic stimulation and increases after an omission, further indicating that the mouse monitors recent stimulus statistics. Together, this provides challenging insights for improving our understanding and theories of predictive processing, highlighting the promise of fast and bright, large-scale imaging.

## Results

We investigated the neural representation of unexpected stimulus omissions in extended sound sequences across multiple spatial scales and cortical depths of the auditory cortex (AC) in mice (CBA/JRj, female, age: 2–3 months, n = 10, Fig. 1A). We locally transfected these normal-hearing mice with the fast and bright Ca²⁺-indicator jGCaMP8m (Fig. 1B) and chronically imaged the whole auditory cortex (1p widefield, 100 Hz) and multiple fields of view (FOVs, 500 × 500 µm) at cellular resolution across different depths (2p resonant scanning, 30 Hz). Imaging was performed through a cranial window (ø = 3.0–4.2 mm) centered above the AC starting after 2–3 weeks of viral expression. Facial motion and pupil diameter were recorded and served as a fast behavioral readout of the acoustic stimulation and the omitted stimuli. Details on statistical testing are collected in Supplementary Data 1.

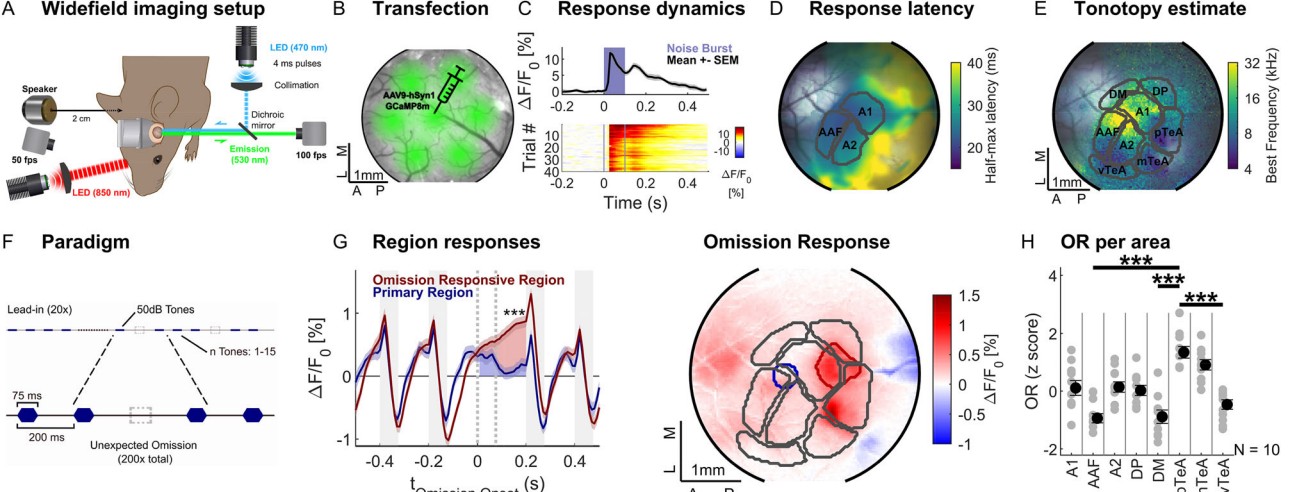

**Fig. 1 | Auditory omission responses consistently localize to the temporal association area. A** Widefield calcium imaging was performed from the auditory cortex in awake, head-fixed mice at a sampling rate of 100 Hz. **B** The fast and bright calcium indicator jGCaMP8m was expressed by injecting the viral construct in 7 locations across the auditory cortex (triangular pattern, 1 mm separation) to achieve rather uniform expression. **C** jGCaMP8m enables clear separation of the onset and offset responses (average ΔF/F₀ over last 10 trials, top) even on a single-trial basis (bottom), here in response to brief white noise sounds (100 ms, 50 dB SPL, average over primary region, blue isoline in **H**). **D** jGCaMP8m's fast rise time of ~2.5 ms allows a response latency map to be estimated from widefield recordings, which shows a low-latency core, composed of A1, A2, and AAF (division shown here based on tonotopy, see **E**). Stimulus as in (**C**); transparency adapted based on overall response size. **E** The overall division of the auditory cortex into subfields was based on the low-latency center and further subdivided based on classical tonotopic gradient reversals. This revealed both primary (AI, AII, AAF) and secondary areas (DM, DP), including the temporal association area, subdivided

spatially into vTeA, mTeA, and pTeA (see "Methods" section[36,37]). **F** We presented sequences of identical sounds which were unexpectedly interrupted by omissions, i.e., a stimulus that was randomly left out. In different trials, the sounds ("standards") were either pure tones or white noise (50 dB SPL, 75 ms, 125 ms pause). Overall, the sequence contained 1440 sounds and 200 omissions (see "Methods" section for details). **G** The neural activity following an omission (left) differed across the auditory cortex (right), with high activity observed in more posterior regions (referred to as the 'omission-responsive region' ORR, red outline) in comparison to the low-latency primary regions (PR, blue outline). The omission response (OR) showed an inflection around omission onset and remained elevated until the response to the post-omission stimulus (PR & ORR defined as adjacent pixels with high stimulus response or OR, respectively, area estimate overlaid in gray).
**H** Across subjects, pTeA showed significantly higher OR than vTeA (p = 0.0001), DM (p < 0.0001), and AAF (p = 0.001, Kruskal–Wallis test after Bonferroni correction, N = 10, OR z-scored within subjects, error bars indicate SEM). Source data are provided with this paper.

## Omission responses originate from the temporal association area posterior to the auditory cortex

We have previously demonstrated the existence of neurons in the AC that respond to the omission of an expected sound using electrophysiological recordings[25]. However, many crucial aspects remained uncertain, in particular, whether these neurons are locally grouped and whether they reside in particular cortical layers. Calcium imaging with jGCaMP8m[35] provides a highly suitable method to address these questions, as it fuses fast dynamics (Fig. 1C top), high signal-to-noise (Fig. 1C bottom; single-trial responses to sounds), with a direct visualization of cellular locations.

We first mapped the size of omission responses across the AC using widefield imaging. Division of the AC into classically described subareas[36,38–40] was performed using a combination of response latency and stimulus modulation to demarcate core areas, and tonotopy to arrive at subareas (Fig. 1D, E, see "Methods" section and Fig. S3 for details). The low-latency core area was further divided into areas A1, A2, and AAF, while the surrounding longer-latency areas were divided into fields DP, DM, and TeA (temporal association area)[41]. TeA has recently been highlighted as a secondary area with suspected top-down influences[37], and we divided it into three anatomical parts (ventral, medial, and posterior TeA).

The acoustic stimulus was a long sequence of regularly spaced (200 ms between stimulus onsets, i.e., Stimulus Onset Asynchrony; SOA), identical sounds, of which a random subset was omitted (Fig. 1F, see "Methods" section for details). As before, the Omission Response (OR) was defined as the mean activity from omission onset to the subsequent stimulus onset (Fig. 1G left, shaded areas[25]). The area that exhibited the largest OR was located postero-lateral to the primary areas (Fig. 1G right), and here the neural activity exhibited a substantial increase at the beginning of the omission and increased further from there (Fig. 1G left, red), sometimes plateauing, similar to what we observed previously using electrophysiology[25], even though magnitude estimates depend on baseline choice during continuous stimulation (See Supplementary Fig. 7D, F). In contrast, the primary area (Fig. 1G right, blue circle) showed a much smaller or absent omission response (Fig. 1G right, blue). The dynamics of the omission response are markedly different from those of offset responses, which exhibited a faster decay (~100 ms, compare to Fig. 1C top). The posterior and medial parts of TeA exhibited the strongest OR, which was significantly greater than in other areas of AC (Fig. 1H, $p < 0.001$, Kruskal–Wallis with Bonferroni correction for multiple testing, $n = 10$, see Fig. S1 for omission and stimulus responses in all animals). Omission responses were observed for different frequencies and also noise bursts and showed overlapping but non-identical spatial profiles, without clear tonotopic organization across animals (Supplementary Fig. 2).

## Omission and stimulus response sizes differ strongly in neurons from different areas

Next, we addressed the relation between responses to (predictable) sounds and an omitted sound for individual neurons in different locations of the AC. Using 2-photon imaging (Fig. 2A), we acquired multiple FOVs targeted to particular locations of interest determined from widefield imaging in the same animal (Fig. 2B), specifically the omission-responsive region (ORR, Fig. 2B red) and the low-latency region with primary-like responses (PR, Fig. 2B blue).

Cells in the ORR FOVs showed strongly elevated activity in the omission period (Fig. 2C top), corresponding to the widefield recordings (Fig. 1G), rising at or around the start of the omission period. Cells in the low-latency, primary region further anterior exhibited much reduced responses in the omission period, and typically larger evoked responses to the stimulus (Fig. 2C bottom).

The distribution of individual cells with dominant responses to omissions (Fig. 2D, dots) mirrored the spatial pattern of widefield responses (color-coded background); however, the transition was

steeper in 2p (Fig. 2E, orange vs red), as expected based on light-spread in the tissue[42]. The OR was also substantially larger (~3×) than the stimulus response in the ORR (Fig. 2F left, $p < 10^{-10}$, linear mixed-effects model, group effect, animal as random factor, $n = 5$ mice). In the PR, the stimulus response was significantly larger than the omission response (Fig. 2F right, $p < 10^{-10}$, as above). As the stimulus response is within a long sequence of sounds, it was already strongly adapted in the primary region. A much greater fraction of cells showed a significantly positive OR in the ORR than in the PR (96% vs. 27%, Wilcoxon signed-rank test at $p < 0.05$). Cells in PR showed largely no significant change in activity (71%), while only a small fraction showed a significant decrease (2%). Comparing the two responses between the regions shows a significant separation (Fig. 2H; ORR: red vs PR: blue, group differences were tested separately for OR and SR, both $p < 10^{-10}$, method as above). This difference can be summarized with the Omission Index OMI := (OR-SR)/(OR + SR), which showed a substantial and significant difference between ORR and PR (Fig. 2I, $p < 10^{-10}$, as above).

Taken together, these results indicate that the OR constitutes a strong, population-wide modulation of activity in TeA, while cells in primary areas do not strongly respond to the omissions.

## Omission responses are most prominent in granular and supragranular layers

Theories of predictive coding typically include a hierarchical organization of prediction-related activity at multiple spatial scales[2,4,6]. On the local level, cortical circuits have been demonstrated to include hierarchical differentiation, e.g., receiving top-down input via L1 and bottom-up input via L4, while sending interregional output via L5[43]. Recent research in the visual system has provided evidence for a separation of prediction-error related signals in L2/3 in the case of deviant stimuli[44]. Here, we investigated whether the more prediction-related omission responses are also more prominent in superficial compared with deeper layers (Fig. 3).

We acquired 2p depth stacks in FOVs located in either primary or omission-responsive region ranging from the brain surface (~50 μm) down to the middle of L5 (~500 μm[45],) (Fig. 3A). In the omission-responsive region (ORR), omission responses were more prominent in superficial layers (Fig. 3B, top) and substantially reduced with depth ($p < 10^{-50}$, Kruskal–Wallis test). There appeared to be two separate steps, one between 100 μm and 200 μm (L2–L3) and one between 300 μm and 400 μm (L4–L5). Stimulus responses also reduced significantly with depth; however, only slightly in absolute terms, given their small overall size. By contrast in the primary area, omission and stimulus responses exhibited a very different depth dependence, with the stimulus response increasing strongly from the top 100 μm to 200 μm (Fig. 3B, bottom, Fig. 3D). While overall much lower than in the ORR, the omission response here decreased first significantly from L1/2, reaching negative values in L3 (200 μm), and then increased again in L4/5 ($p < 0.001$, Kruskal–Wallis test, with Bonferroni-corrected post-hoc testing, see Fig. 3C for details). The stimulus response was instead rather low in L1/2 and increased significantly to L3 and deeper layers.

Overall, omission and standard responses showed substantial dependency across depth, which further depended on the region of the auditory cortex. The layer profile of ORs in TeA appears consistent with top-down inputs as proposed previously[4].

## Pupil dilation and brief facial movements are indicative of sequence statistics

While the animals were not performing a task, we found their innate behavioral responses to be indicative of monitoring the sequence structure. We analyzed the pupil diameter and facial movements from video recordings (Fig. 4A, B). During omissions, the pupil diameter increased significantly (Fig. 4C, $p = 0.002$, Kruskal–Wallis ANOVA, $n = 8$ mice), indicating heightened arousal[46]. Although pupil responses are

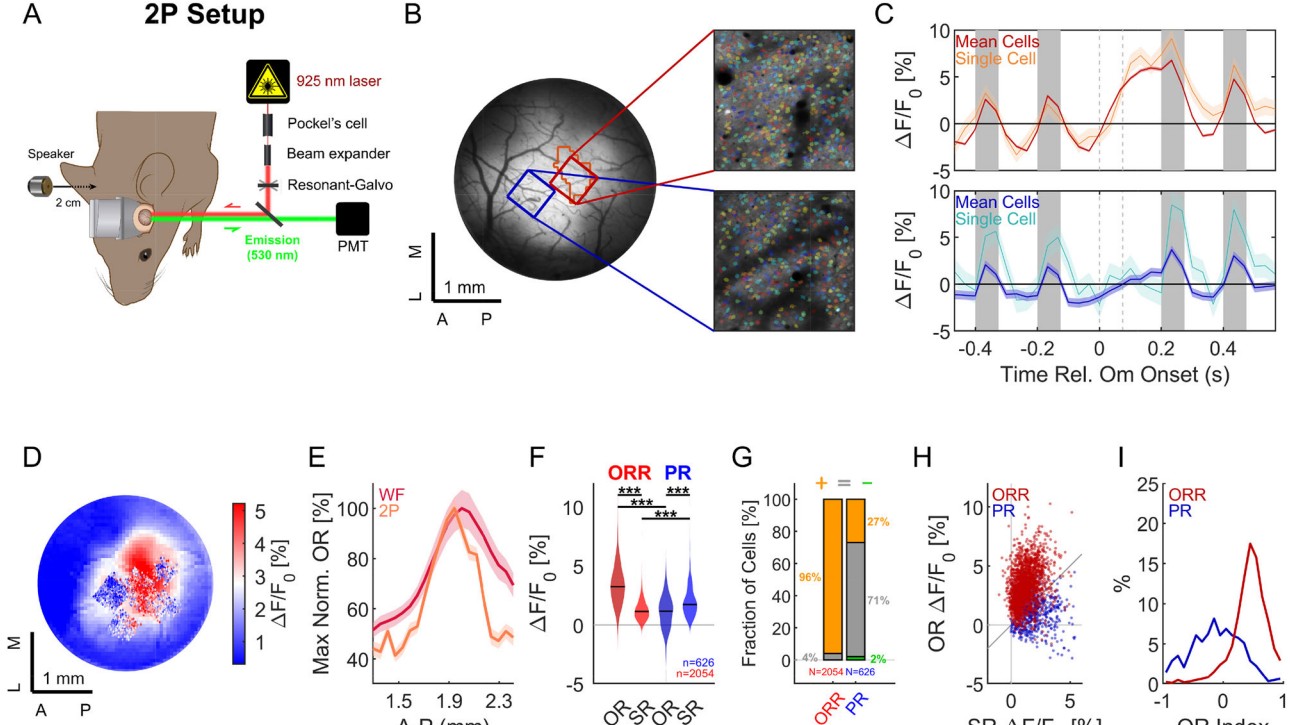

**Fig. 2 | Omission responses in single cells dominate in the omission-responsive region. A** Two-photon (2p) imaging was performed in the same mice based on the widefield-based mapping of the auditory cortex (Fig. 1). Cells were registered into widefield space relative to a 3D origin that was re-established every imaging session by use of a vessel landmark. **B** Multiple fields of view were collected in 2p from the primary (blue) and omission-responsive (red) area based on the corresponding widefield image. Colored dots (right plots) indicate cells detected using Suite2P. **C** In the primary region, the responses to sounds exceeded the activity during and after the omission period (dashed lines). In the omission response region, sound responses tended to be smaller; however, during the omission period, a strong, positive response occurred (dark colors: FOV average, light colors: representative single cell). Traces show mean; shading is ± SEM. Same for (**E**). **D** Cells with strong omission responses (dots, color scaled to 2p OR) occurred predominantly in locations consistent with the widefield OR (also color scaled). **E** The ORR boundary was more clearly defined in 2p (orange) than in widefield (red), suggesting that in

widefield, non-somatic contributions and light dispersion broaden the signal spatially. **F** In the ORR (left), the OR was significantly larger than the standard response (SR) in single cells. Conversely, in the PR (right), the SR was significantly larger than the PR (cells collected from 24 FOVs in 5 animals, same for the following panels, see "Results" section and "Methods" section for details on statistics). Asterisks denote statistical significance: $p < 0.05$ (*), $p < 0.01$ (**), $p < 0.001$ (***). **G** In the ORR, a much greater fraction of cells (96%) showed a significantly positive OR than in the PR (27%), where most cells showed no significant change (71%), and a small fraction reduced their activity (2%). **H** In the ORR, the response to the omission was substantially larger than to the stimulus. Conversely, in the PR, the stimulus response was larger than the omission response. **I** Cells in the ORR had a substantially larger Omission Response Index ORI ($p < 10^{-10}$, statistics as in **F**), defined as OR-PR/(OR + PR), than cells in the primary region. Source data are provided with this paper.

generally slow[47,48], the dilation of the pupil becomes significant 110 ms after omission onset ($p = 0.027$, Wilcoxon test), already before the presentation of the next stimulus. Facial movements have recently been demonstrated as a basic readout of an animal's reception of visual and acoustic stimuli[49], with subcortical origin but under cortical modulation[47]. Facial movements, in particular of the whisker pad (Fig. 4A, right), occurred closely timed after stimulus onset ($\Delta t$-60–70 ms) and lasted only briefly (-100 ms, Fig. 4D). This induced facial motion was significantly larger after an omission than before (Fig. 4E, $p < 0.01$, Wilcoxon rank sum test). Subsequently, the response adapted within 5 stimulus repetitions (Fig. 4F, $p < 0.001$, Kruskal–Wallis test). Conversely, the post-omission stimulus response depended significantly on the number of consecutive sounds that preceded the omission (Fig. 4G, $p < 0.001$, Kruskal–Wallis test). Adaptation of stimulus response magnitude was similar in the neural data (see also ref. 25), the reduction being stronger in the ORR (57%) as opposed to PR (23%). Additionally, neural responses and facial movements were positively correlated across stimulus representations within 4 stimuli after the omission, with the ORR showing a higher degree of correlation than the PR (Pearson $r = 0.229$ and 0.116, respectively, both $p < 0.001$). Together, these results are indicative of the mouse monitoring acoustic sequence statistics beyond a basic

reflex on the neural and behavioral level, even without training or reward.

## Omission responses cannot be explained as offset responses
Neurons in the auditory cortex respond not only to the onset and spectrotemporal features of ongoing sounds, but also to the offset of sounds (see, e.g., Fig. 1C[50,51]). While offset responses had a much faster time-course than the omission responses (compare Fig. 1C/G and Supplementary Fig. 10) and did not share the same map across the auditory cortex (compare Fig. S3D, E and Fig. 1H), we ran multiple controls to rule out the possibility of mistaking offset responses or refractory characteristics for omission responses (Fig. 5). We reasoned that true omission responses must only be present when a sound is expected, exhibit a time-course that starts at the expected stimulus onset and be robust to SOA and frequency content.

The first control was a comparison of the neural activity in the omission period in 200 ms SOA sequences with the interstimulus (non-omission) period in 400 ms SOA (Fig. 5A). In the latter sequence, a long pause is expected between the stimuli, while in the former, a pause of this length between the same stimuli is an unexpected deviation. In both cases, stimulus-driven offset responses should be similar, and we can therefore attribute differences between the neural activity in these

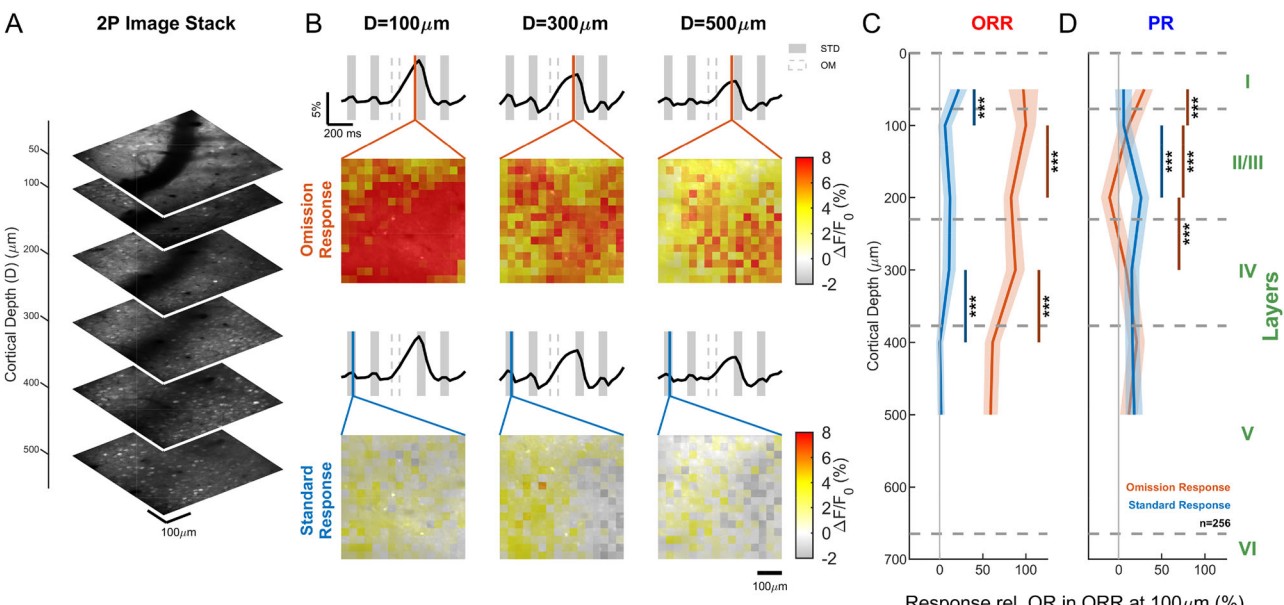

**Fig. 3 | Omission responses are most prominent in superficial layers of the omission response region. A** Mean two-photon fluorescence images displayed as stacked slices at increasing cortical depths. Each image corresponds to a 512 × 512 pixel field of view acquired at the same XY location and shown at its respective depth (50, 100, 200, 300, 400, and 500 μm). Images are displayed in grayscale, with the stack providing a 3D representation of fluorescence across cortical depth. **B** In the omission-responsive region, the omission response (OR, time-point of area plot in second row indicated by red line, top rows) reduces from more superficial cortical layers to deeper layers (left to right). The response to the standard tone in the omission-responsive region (SR, indicated by the blue line, bottom rows) remained similar in total size as a function of depth. Data shows locally averaged calcium activity from a depth stack from a single mouse (m216). **C** In the omission-responsive region, the average omission response (red) decreased significantly

with depth, with particularly substantial decreases from L1 to L2 (-20%) and from L4 to L5 (-30%). The standard response (SR, blue) was substantially smaller in size, but also showed significant variation with depth, first decreasing from L1 to L2, and then increasing again to L3. Data from depth stacks in 2 mice. OR and SR are both quantified as the average response over show time windows (OR: from 0 ms to 200 ms after omission onset; SR: from 0 ms to 100 ms after standard stimulus onset). **D** In the primary region, the omission response was substantially smaller (-81%) across depths ($p = 0.002$, Wilcoxon rank sum test). Shaded areas in C/D show ±1 SEM, where N is corrected by the average correlation of pairs as follows: $N_{eff} = N/(1 + (1-N)*C)$, where C is the average correlation between the activity at different locations[117]. Layer boundaries drawn in accordance with[45]. Significance: $p < 0.05$ (*), $p < 0.01$ (**), $p < 0.001$ (***) (see Supplementary Data 1 for statistical details). Source data are provided with this paper.

cases to the response to the omission. In the Omission-Responsive Region (ORR), the omission response in the SOA 200 ms sequence (OR 200, Fig. 5B, red & light gray) was much more pronounced than the interstimulus response in the 400 ms SOA sequence (ISR 400, orange & dark gray). The difference between the two localized again to the ORR (Fig. 5C), and this difference was highly significant within an animal across within-sequence repetitions (Fig. 5D, $p < 10^{-10}$, Wilcoxon Rank Sum test, $n_1 = 162$, $n_2 = 1428$) and across animals (Fig. 5E, $p = 0.0079$, Wilcoxon Rank Sum test, $n = 5$), as well as in single cells recorded in the ORR in 2p in the same animals (Fig. 5F, $p < 10^{-100}$, Wilcoxon Rank Sum test, $n = 765$).

The second control was the presentation of two subsequent omissions. These formed only a small subset of the total number of omissions (10/162), in order to not accustom the animal to this length of omission or strongly interfere with the predictability of the standard sequence. In addition to the first OR, we also observed a second, time-locked increase in the neural activity after the beginning of the second omission (Fig. 5H), measured by the difference between the second and first OR times the first OR size (double Omission index, dOM, see "Methods" section). The cortical region where this second omission occurred was qualitatively similar to the one in response to the first omission (Fig. 5I, J), while further extending to more posterior regions. The further increase for the second OR was consistent across all five animals in which this condition was run (Fig. 5K). To our knowledge, no human study exists which included double omissions and collected neural recordings. We are therefore not sure whether this would also be observed in humans in addition to the first omission response (see "Discussion" section).

The third control was the use of multiple SOAs to investigate the timing of the resulting omission responses, ranging from 100 to

400 ms (Fig. 5L). From human[20] and animal studies[25], we expected to find reduced neural responses to unexpected omission as the SOA increased. Consistently, we found that the size of the OR depended significantly on SOA (Fig. 5O, $p = 0.009$, Kruskal–Wallis test), and the slope reduced significantly with SOA (Fig. 5P, $p = 0.0011$, Kruskal–Wallis test) while the minimum value of the SR decreased significantly from SOA = 100 ms to SOA = 200 ms and longer (Fig. 5Q, $p = 0.01$, Kruskal–Wallis test; $p = 0.001$, Wilcoxon rank sum test between 100 ms and all longer SOAs grouped).

Altogether, the omission responses appear to be genuinely related to an expectation of the omitted sound, as they differ in their dynamics from offset responses (see also Fig. 1C, S3D, E, and Supplementary Fig. 10), occur also for a second omission, and change in time and size depending on the sequence structure.

**Omission responses partially scale with the level of regularity in the stimulus sequence**

If the OR reflects a prediction in a regular sequence of stimuli, then its size would be expected to scale with the degree of regularity in the sequence[52–54]. We assessed this question using two different controls, a temporally jittered version of the paradigm and an analysis of subsections of the stimulus sequence.

The jittered sequence was designed to be on many levels comparable to the standard SOA 200 sequence (Fig. 6A), i.e., it had the same average SOA of 200 ms, same standard sound (white noise), same order of sounds and omissions, same sequence length, while only the precise timing of the sounds was modified randomly in a balanced way, i.e., the SOA following a sound was randomly increased or decreased by up to 100 ms (in steps of 50 ms). The pseudorandom

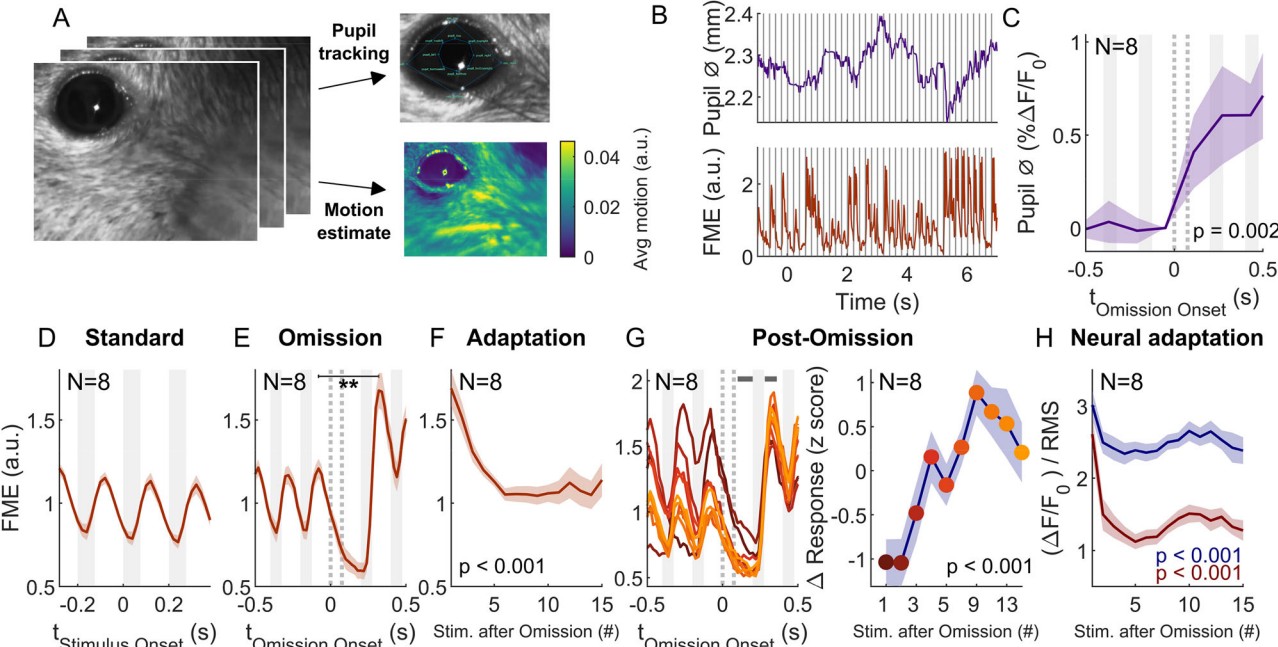

**Fig. 4 | Pupil dilation and brief facial movements are indicative of sequence statistics. A** During widefield imaging, the right facial area was recorded under infrared illumination, including the eye and the whisker pad. Facial motion was mostly driven by the whisker pad on the mouse's snout, here given as average motion per pixel. **B** The pupil diameter (ø) was tracked using SLEAP (top[114], while facial motion components were analyzed by evaluating the different components underlying frame-to-frame changes in the image (bottom[47]. For the latter, we selected the singular value decomposition component with the highest variance when aligned to the stimulus onsets. This component showed fast positive transients time-locked to the occurrence of sounds. Pupil diameter (ø) changed more slowly overall with some intermittent transients, which we attributed largely to noise, e.g., by whiskers moving in front of the eyes or pixel jitter in the pupil estimate. **C** Pupil diameter increased significantly around the time of omission ($p = 0.002$, Kruskal–Wallis test, downsampled from 50 Hz to 6.25 Hz, baselined to a linear extrapolation of the pre-omission diameter. **D** Facial motion energy was closely time-locked to each sound's onset at a latency of -65 ms and lasted only for

-100 ms. FME is aligned to sound onset at $t = 0$ preceded by at least 5 sounds (to avoid influence of preceding sound omissions), averaged over all stimulus presentations from all animals ($N = 320$ per animal). Error hull indicates 1 SEM across animals in all panels. **E** FME dropped during an omission, but subsequent sounds elicited substantially greater movement, compared to the pre-omission sounds ($p = 0.0047$, Wilcoxon rank sum test). **F** FME in response to tones after an omission showed a significant decrease for positions later in the sound sequence ($p < 0.001$, Kruskal–Wallis test). **G** FME increased significantly with the number of stimuli preceding the omission ($p < 0.001$, Kruskal–Wallis test, difference between the time ranges indicated by gray bars in the left panel), i.e., reduced more during the omission and increased more after it for longer pre-omission stimulus sequences (left). Colors correspond to different positions after omission (see right panel). **H** Analogous to the adaptation of facial motion (**F**), neural tone responses in PR (blue) and ORR (red) decreased significantly with the number of standards ($p < 0.001$, Kruskal–Wallis test). The ORR showed a stronger adaptation than the PR (−57% and −23%, respectively). Source data are provided with this paper.

sequence of temporal shifts was identical after omissions, which made it completely random before the omissions (due to the random number of standards in between omissions). In this way, we could also inspect the adaptation of the aligned sound responses after the omission (Fig. 6B).

We expected a reduced OR, due to the reduced temporal predictability; however, not a complete abolishment of the OR, as there remained a predictability in sound identity and an average predictability in time. This hypothesis was confirmed by the data in both widefield (Fig. 6B–E, maroon vs. ocher) and cell responses (Fig. 6F), which showed significant reductions for JIT 200 compared with SOA 200 both within an animal (Fig. 6D, $p = 0.0014$, Wilcoxon signed-rank test) and across animals (Fig. 6E, F, WF: $p = 0.03$, Wilcoxon signed-rank test, $n = 5$ mice; 2 P: $p < 10^{-100}$, Wilcoxon signed-rank test, $n = 736$ cells). Spatially, the difference between the two measures was strongest in the omission area (Fig. 6C).

Secondly, we analyzed whether the number of stimuli in the sequence had an effect on the OR, both on shorter and longer time-scales. To analyze shorter time-scales, we binned the omissions by the number of preceding standards, $N_{STD}$ (Fig. 6G), with each $N_{STD}$ occurring equally often. We find only a weak dependence on $N_{STD}$, showing an increase in OR for both WF and 2 P (Fig. 6I, J). However, in both cases, referencing the post-stimulus trough preceding the omission period made this dependence insignificant, which could point to a baseline shift as a function of $N_{STD}$.

Next, we analyzed the longer time dependence on sequence exposure. We find that as a function of the position of the omission in the overall sequence, the OR first increased and then slightly decayed again (Fig. 6K, early: light blue; late: dark blue; $p < 0.008$, Wilcoxon rank sum test, $n = 5$ mice). While the division point for this early/late trend analysis was chosen manually (here: 60 trials, where trial refers to an omission), the result was not strongly dependent on the choice, with divisions in the range of 40–70 trials giving similar results.

We also wondered whether the omission response could be a consequence of entrainment to the repeating stimulus (Fig. S9[55–57]). We first analyzed whether the stimulus response was entrained to the sequence, i.e., whether the response size (measured as peak-to-trough) increased as a function of time (see "Methods" section). Stimulus entrainment was observed only in the ORR (Fig. S9F), and only after accounting for early, positively dominated responses, which adapted over the same time scale as the entrainment increased (compare Fig. S9C to Supplementary Fig. 9F), estimated by fitting an envelope to the response (Fig. S9A, B). The omission response did not show entrainment itself, i.e., its shape stayed very different (slow, unipolar) from the stimulus response (rapid, bipolar), and consequently the entrainment measure remained very low (Fig. S9G, H).

In summary, the size of the omission response was sensitive to the regularity of the sequence on the level of the temporal spacing of the stimuli, as shown in the jitter control (Fig. 6F), as well as on the degree to which the regularity has been experienced, as shown by the

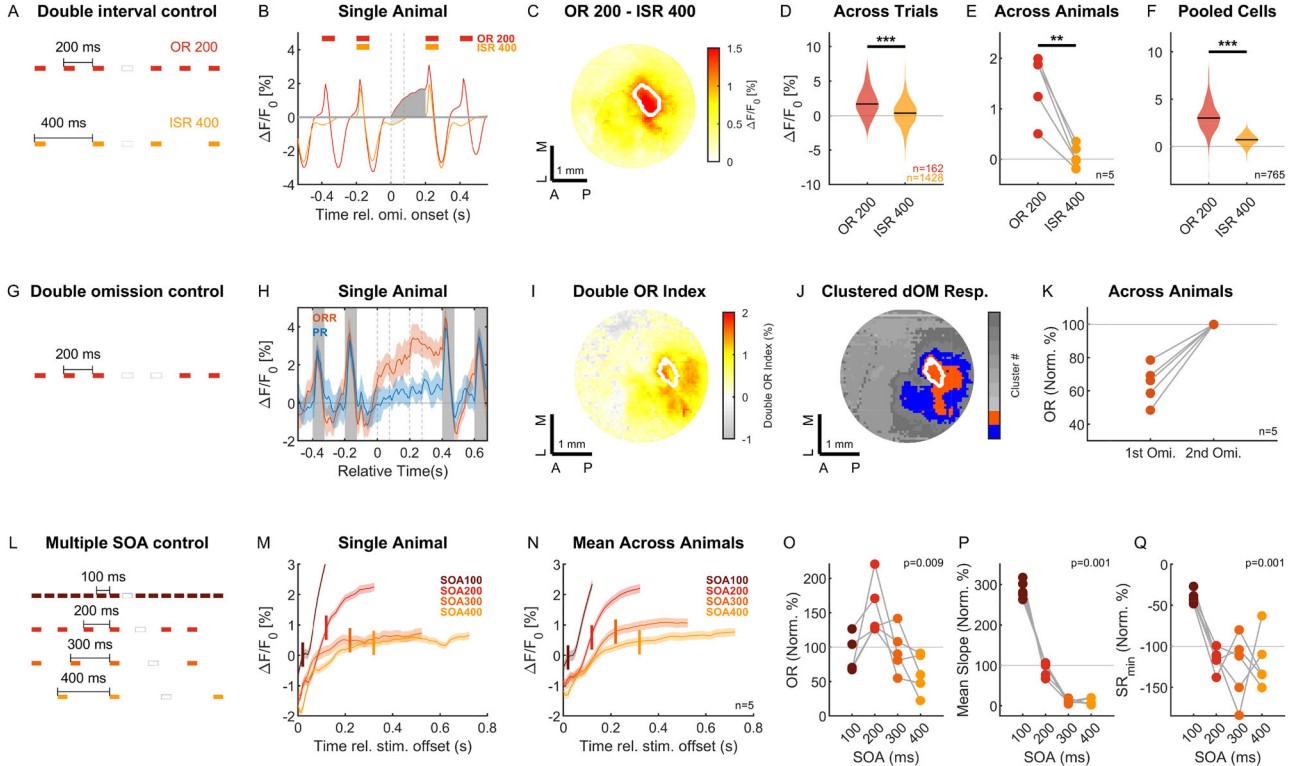

**Fig. 5 | Omission responses are not explained by stimulus-driven response dynamics. A** We compared neural responses to omissions in a standard sequence with an SOA of 200 ms to *non-omissions* in a sequence with an SOA of 400 ms. The identical pause between stimuli is the normal, expected interval in SOA 400, but is an unexpected deviation in SOA 200, and therefore controls for basic stimulus-driven response dynamics, such as offset responses. **B** SOA 200 omissions responses (OR, light gray) were substantially larger than SOA 400 interstimulus responses (ISR, dark gray). **C** The difference between OR 200 and ISR 400 localized to a very similar area as the OR 200 itself. Across trials (**D**) in the animal shown in (**C**), as well as across animals (**E**) and cells from the ORR of all recorded animals (**F**), the same behavior was observed. Significance: $p < 0.05$ (*), $p < 0.01$ (**), $p < 0.001$ (***) (see Supplementary Data 1 for statistical details). **G** We checked whether a second omission in the SOA 200 sequence led to a further increase in the neural response by introducing double omissions into the sequence of standard tones. **H** Comparing the primary region (PR) with the omission-responsive region, a timed increase in neural activity occurs at the onset of the second omission on top of the elevated response from the first omission for a representative animal. Mean with ±SEM shading. Same for (**M** and **N**). **I** An index of the additional increase to the second omission localized as well to the omission area, i.e., the medial and posterior parts of TeA. **J** Cluster analysis of the second omission response confirms a location consistent with the ORR, while largely absent in the PR. **K** The double omission paradigm was run in 5 animals, each of which showed a further increase in response to the second omission relative to the first, normalized to the second omission response size for clarity. **L** We presented multiple SOAs, ranging from 100–400 ms, to investigate the dependence of the response dynamics on the time of expected occurrence of the next stimulus. **M, N** The OR showed different time-courses depending on the SOA, faster for short SOAs, slower for longer SOAs, in single animals, as well as the average across animals (**N**, $n = 5$). **O** The OR showed a bandpass-like dependence on SOA, similar to ORs in previous studies in humans, rats, and mice[20,25,26]. **P** The average slope of the OR significantly decreased with longer SOAs ($p = 0.0011$, Kruskal–Wallis test). **Q** The minimum value of SR shows a significant decrease from SOA = 100 ms to longer SOAs ($p = 0.001$, Wilcoxon rank sum test). Source data are provided with this paper.

---

omission response over the entire duration of the sequence (Fig. 6K, as in ref. 26). On the other hand, the dependence on the more recent regularity, i.e., the number of standard stimuli between omissions, was weak and appeared to be already included in a baseline shift. Together, this suggests that omission responses are regularity dependent, but their build-up extends beyond the longest inter-omission sequence (15 tones) used here.

**Omission responses are spatially distinct from prediction errors and stimulus-specific adaptation**

Previous research into predictive processes in the auditory system often investigated neural responses to unexpected changes in stimulus properties[13,16,30,58]. The classical paradigm in this context is regular sequences of tones, with randomly occurring oddball/deviant stimuli interspersed. Previous research has found that the deviant response increases with the stage of the auditory system from the inferior colliculus to the thalamus and the auditory cortex[14,19]. We asked next how the spatial organization of omission responses related to those attributed to deviant stimuli and prediction errors. To quantify these in accordance with the literature, we ran two classical paradigms (Fig. 7A,

left), oddball/deviant sequences (Dev), and a Multiple Standards Control sequence (MSC). In the Dev case, we replaced the omissions with deviant sounds (4 and 32 kHz as standard or deviant, SOA 200 ms), while in the MSC case, the omissions were replaced by standard stimuli (4 and 32 kHz), and the standards were replaced by pseudo-randomly drawn tones of different frequencies.

In accordance with previous studies, we defined two measures: *Repetition suppression* (*RS*) as the difference between the neural responses to the same tone when it is a standard in the *Dev* case and when it is in the context of random tones inside the MSC (Fig. 7A, right), averaged for both probe frequencies. The rationale is that repeated presentation of the same stimulus leads to a reduction in the response size, be it due to any combination of active suppression or passive adaptation. The response to the same stimulus in the MSC, where it is not preceded by itself, is considered a (rather) unadapted reference in comparison.

Secondly, *prediction error (PE)* was defined as the difference between the neural response to the same tone when it is a deviant in the *Dev* case and again in the MSC (Fig. 7A, right). The rationale is that when it is a deviant, the same tone is less expected, and the response could

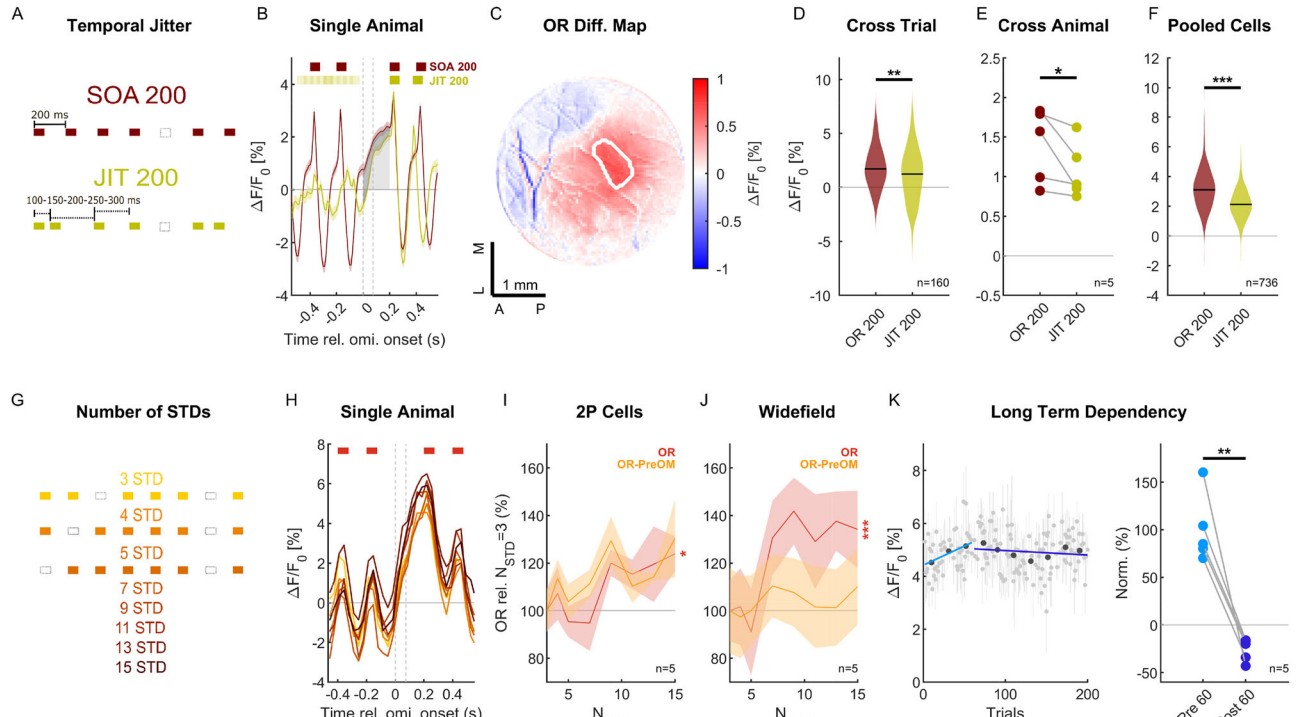

**Fig. 6 | Omission responses scale with the level of regularity in the stimulus sequence. A** We compared the standard SOA 200 omission sequence (maroon) with a jittered version (JIT 200, ocre), in which omissions occurred at the same positions, but the SOA after each tone was pseudo-randomly shortened or lengthened by up to 100 ms (in steps of 50 ms). This reduced the temporal predictability in the sequence, while maintaining spectral predictability. The omission period was kept at 200 ms to compare with SOA 200. **B** The OR in JIT 200 (light gray) was reduced in comparison to SOA 200 (dark gray), here shown for a widefield recording in a single mouse, averaged over the omission-responsive region (ORR). Mean with ±SEM shading. Applies to (**I** and **J**). **C** The difference between the OR in SOA 200 and JIT 200 was largest in the ORR (white outline); however, also showed a difference in surrounding and more anterior areas. **D** Jittered sequences produced significantly smaller ORs than standard sequences in the mouse shown ($p = 0.0014$, Wilcoxon signed-rank test, $n = 160$ trials). **E** This difference was consistent across all animals in which this paradigm was studied ($p = 0.03$, one-sided Wilcoxon signed-rank test, $n = 5$). **F** The same difference was observed on the level of cells in 2p imaging in the same animals in the ORR, pooled across all animals ($p < 10^{-100}$, Wilcoxon signed-rank test, $n = 736$ cells). **G** Sequential predictability was expected to increase with the number of standard tones ($N_{STD}$) preceding an

omission. We sorted all omissions in our sequence according to $N_{STD}$ and then analyzed the OR in each case (Colors indicate different $N_{STD}$). **H** Single-cell responses showed a tendency towards larger ORs as a function of $N_{STD}$ in a representative animal (Colors, see **G**). **I** Across animals, the cell's OR showed borderline significant dependence on $N_{STD}$ (red, $p = 0.02$, Wilcoxon Rank sum test, $n = 5$) when comparing two $N_{STD}$ ranges ([3–7] vs [9–15]). However, when referencing the preceding trough, this comparison was not significant (orange, $p = 0.16$). **J** In the widefield recordings, the OR increased significantly with $N_{STD}$ (red, $p = 7 \times 10^{-4}$, Kruskal–Wallis test); however, when normalized for the pre-omission period, this increase again disappeared (orange), suggesting a shift of the response already before the omission as a function of $N_{STD}$. **K** Over the whole duration of the stimulus sequence (200 omissions, 1440 tones) the OR initially increased, before weakly decreasing again, indicated by the slopes of two linear fits (light blue: trials 1–60; dark blue: trials 60–200; gray dots are average ORs across 5 mice ± SEM, black dots are averages over 16 consecutive trials). Across all animals, the slope of the OR across the early trials (light blue) was positive and significantly greater ($p < 0.008$, Wilcoxon rank sum test, $n = 5$) than the slope across later trials (dark blue), which was negative. Source data are provided with this paper.

therefore represent a prediction error. Below, we first inspect the spatial maps of the responses before stepping towards the maps of PE and RS.

Deviant responses (orange) to the same tone were generally larger than standard (blue) or MSC (green) responses in both primary and secondary areas (Fig. 7B). Deviant responses showed a late increase in the response (100–200 ms, in particular in pTeA), consistent with the ramping/elevated response observed in omission responses (Fig. 1G). In the MSC case, the average response was particularly large in A1, A2, and AAF (Fig. 7C, F). The Dev response was overall significantly larger than the MSC response across all regions ($p < 0.001$, Wilcoxon signed-rank test, FDR-corrected using the Benjamini–Hochberg method). The Standard response tended to be smaller than the MSC response, with the largest reductions in A1/A2.

Repetition suppression (RS) showed significant variation across the different cortical areas (Fig. 7G, I, $p = 0.019$, Kruskal–Wallis across areas), with the highest degrees of response suppression seen in A1 and A2, while no parts of TeA showed adaptation to repeated stimulation. Prediction errors were highest in AAF; however, they did not vary significantly overall across regions (Fig. 7H, I, $p = 0.23$).

In summary, we find qualitative differences in the degree of stimulus-specific adaptation across different regions, focused on A1/2, while remaining negligible in other regions, in particular, the main contributor to the ORR, the pTeA. Prediction errors did not show big differences across areas, in line with the widely represented deviant response and the lack of direct relation between Deviant and Omission responses reported earlier[25].

## Discussion

Sensory predictions are ubiquitous in human perception and may even constitute a normative principle to understand the organization of the sensory cortices[59–61]. We have presently investigated to what degree neural responses in the auditory cortex to the omission of expected sounds can be considered as a reflection of predictive processes. We find that the omission responses (OR) are dominantly localized to layers I–IV in a subregion of the temporal association area (TeA) posterior to primary auditory cortex, and that they differ from more basic offset responses in their temporal shape and size. Further, we show that the OR scales with the level of regularity in the predictive sound

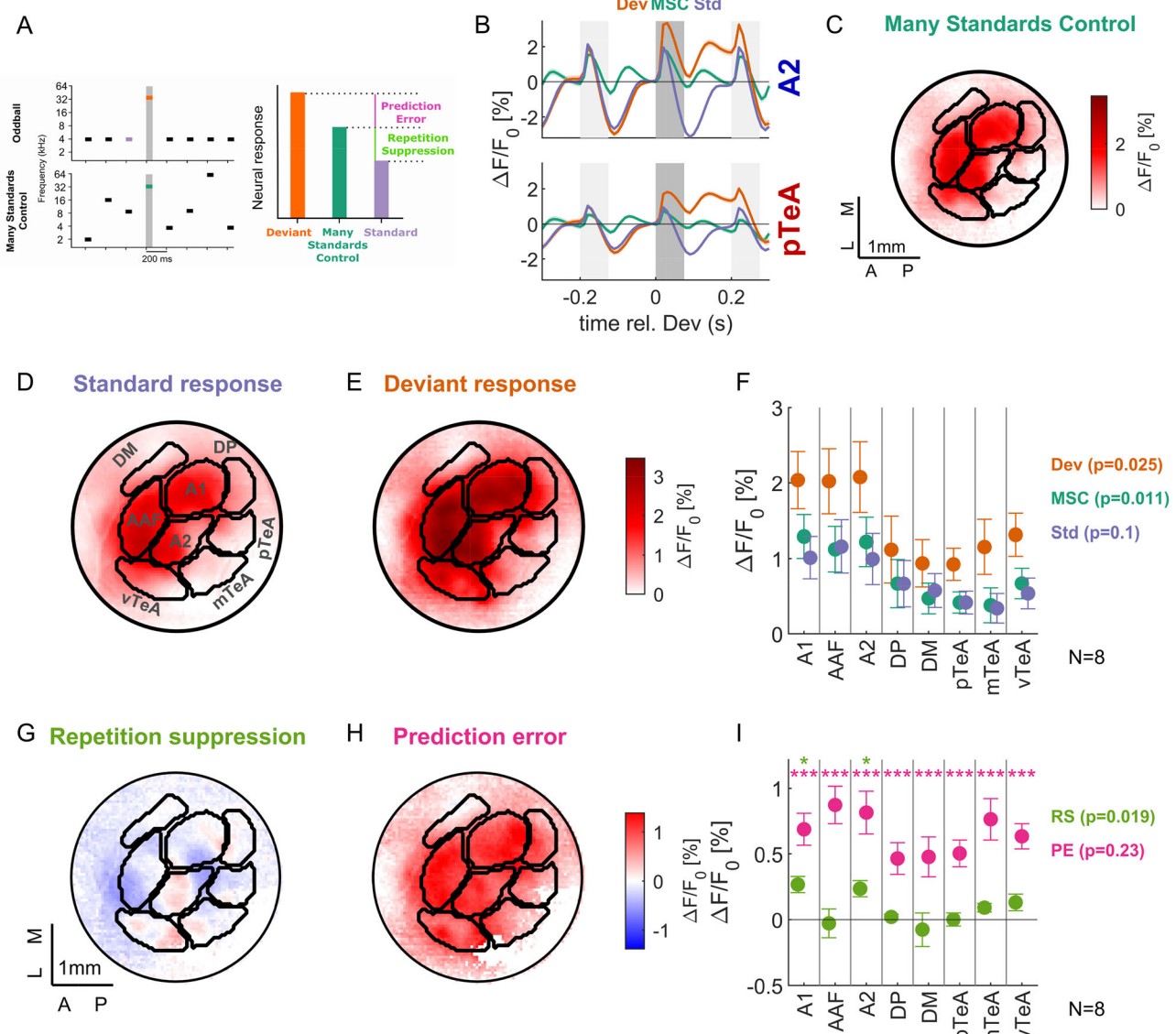

**Fig. 7 | Omission responses localize to different areas than repetition suppression. A** We estimated the degree of response suppression to repeated stimulation and prediction errors using two additional stimulus sequences. The *oddball sequence* (top) was structured identically to the omission sequence, but omissions were replaced with deviant tones (orange). The *many standards control sequence* (MSC, bottom) leaves the oddball sounds in identical positions (green), but randomizes the frequency of the standards. As in previous studies, the prediction error (PE) was the difference between the responses to identical tones when they occurred as deviant or inside the MSC. Repetition suppression (RS) was the difference in response between the tone when it was inside the MSC and when it was a standard in the oddball sequence. To prevent frequency-specific confounds, responses to 4 kHz and 32 kHz were averaged for each condition. We also present the corresponding indices of PE and RS in Supplementary Fig. 6. **B** Responses to the same tone when occurring as a deviant, standard, or in the MSC for the primary region (top) and in the omission-responsive region (bottom). Single mouse average (m193). **C** Responses map to the tone inside the MSC are most dominant in A1, A2, and DP (same animal, response average), omission-responsive area shown as a red line. **D** Response map to the tone when it is the standard in the oddball sequence. AAF shows the least adaptation (same animal, response average). **E** Response map to the tone when it is an oddball/deviant inside the oddball sequence shows high responses widely, with the highest responses in A1, A2, and DP. **F** Population summary ($N = 8$ areas) of responses across areas. A1 and A2 have the highest oddball responses, compared to smaller responses in other areas. Across-area significance was assessed using Kruskal–Wallis ANOVA for the different measures ($p$-values on the right), and significance stars per area indicate $p < 0.05$ (*), $<0.01$ (**), and $<0.001$ (***) (Wilcoxon signed-rank test, with false discovery rate correction via Benjamini–Hochberg); same for (**I**). **G** Repetition suppression of the standard tone is focused on A1 and A2 (red = more suppression), while the omission-responsive area shows comparatively little suppression, i.e., overall negative iRS. **H** The prediction-error index was positive throughout the auditory cortex. **I** Population summary of iRS and iPE across subareas of auditory cortex ($N = 8$ areas), where iRS significantly differed between areas and was strongest in A1/A2, while near 0 in the subareas of TeA. iPE was overall positive, with the highest values in the core areas A1/2/AAF and vTeA. Source data are provided with this paper.

sequence and show a further increase to a second consecutive omission. In relation to current theories of predictive processing (see below), the positive sign of the OR makes it most consistent with a prediction or negative prediction error; however, the temporal shape of the OR is inconsistent with predictions/prediction errors that include the termination of the sound. The present results suggest

modifications to current theories of predictive processing, where the OR might instead constitute an integrated prediction-error signal.

## Relation to previous studies on auditory omission responses

A response to an expected, but missing sound has long been a promising paradigm in the quest to understand predictive processing in

neural systems, as it has the potential to more cleanly separate predictions and prediction errors from accompanying stimulus responses, compared to classical oddball sequences[6].

**Temporal and shape properties.** In humans, responses to omissions were first reported by Yabe et al.[20] and have been confirmed in many studies since, in different variants. Generally, there appears to be a temporal limit up to which basic omission responses are elicited (~300 ms in humans[20]); however, more complex omissions, such as missing stimuli in stimulus groups, are also observed at longer inter-trial intervals[62]. Consistent with the present results, omission responses show a slower time-course than deviant responses[63]. In animals, omission responses have been demonstrated as well using different paradigms. In the auditory cortex, two recent electrophysiology studies demonstrated the existence of omission responses inside oddball sequences[25,26], with varying prevalences (Lao-Rodriguez et al.: ~35%, awake, left AC, mice; ~20% urethane, left AC, mice & rats; Yaron et al.: ~20%, urethane, right AC, rats). In a modified oddball paradigm, Awwad et al. demonstrated responses to omitted brief gaps inside sounds[12]. A number of studies[28,29] have reported responses at the end of slow sequences of sounds (~0.5 Hz), which even repeat multiple times, which have been termed *Echo* responses, which might be based on different principles, but are operationally also omission responses.

The temporal shape of the ORs is consistent with previous electrophysiology studies[25,26], showing an increase closely timed with the onset of the omission, followed by a comparably slow rise until the next stimulus. If the omission is followed by a second omission at the regular interval, another increase is observed, colocalized in AC with the first OR (Fig. 5G–K). At the end of the stimulus sequence, the OR can be seen to stay elevated for a period of 1–2 s (Fig. S7G, H). These results suggest that there is a temporally precise (10 s of ms) mechanism for predicting stimulus timing that underlies the beginning of the OR, while the OR remains elevated even after the expected end of the omitted stimulus. While jGCaMP8m provides substantially improved temporal resolution, the precise timing of predictive computations at the millisecond scale remains to be confirmed with electrophysiology. We show that this response is different from offset responses (Figs. 5 and S10), but decreases in strength with the inter-stimulus interval (Fig. 5), here substantially decreasing at an SOA of 400 ms. While this is consistent with the temporal conditions in humans when ORs have been observed, it does raise the question of whether or where in the brain longer time-scales are encoded. The ramping shape of the OR is unexpected, given most classical accounts of predictive processing, but there might be a connection with evidence integration[64,65], which has been shown to also exhibit ramping behavior with the ramp's slope modulated by the amount of evidence (which in the present case would be the duration of silence in relation to the temporal prediction).

**Localization across brain areas.** A striking and unexpected finding in the present data is the localization of the OR to the temporal association area (TeA), in particular, its medial and posterior parts (Fig. 1H). Importantly, this is different from the localization of the elevated response to deviant stimuli, as shown presently (Fig. 7) and in previous studies[32,66]. It is generally consistent with the localization of omission responses to posterior secondary areas in the rat[25], while in the same study, the localization in the mouse was too imprecise to compare to the present data. TeA has not received a lot of attention in the auditory research in mice, however, it has been shown that TeA is among the main inputs to AC[67], is likely to be a secondary auditory area with long response latency and receiving top-down inputs[37], a higher degree of projections to the amygdala than the AuD[68] and projections to the superior colliculus[69] and plays a particular role in representing pup-calls in mothers[70]. The localization of the omission responses to an area creates the opportunity to study their origin by locally removing top-down input, following similar approaches for oddball deviant responses in the visual domain[44] and in the marmoset auditory cortex[18]. Regarding other brain regions, we have previously shown that - at least in the anesthetized state, omission responses are rare in the inferior colliculus[25], which is consistent with the absence of omission responses in the brainstem response in humans[71]. However, further research is needed to address the existence of omission responses in other subcortical regions and the awake state.

**Localization across cortical layers.** Theories of predictive processing have also suggested different contributions across the cortical microcircuit, i.e., in particular across different depths[4], which we directly addressed using 2p imaging. We find that neurons in the supragranular layers show larger ORs than in the granular layers, with ORs further reducing when going to the infragranular layers. Since no corresponding decrease is observed in the stimulus responses with depth, we conclude that this is not a side-effect of imaging quality as a function of depth. These results are in line with suggestive results about the depth prevalence of OR in previous studies[25,26]. Given the physics of light transmission through tissue[42], this indicates consistency between our widefield and 2p results (see also Figs. 2 and 3).

**Contextual properties.** It has recently been shown that omission responses to global sequence violations are rare[72], although this result might have partly been due to the recording location, which may not have included TeA. Consistently, in the visual system, shifting from local to global pattern violations appeared to shift prediction-related signals to higher cortical stations[73]. Further, as was demonstrated previously, on the time scale of typical omission sequences, independent of whether the omissions are randomly or regularly selected in the sequence, ORs in humans, mice, and rats occur at the same strength[25,71,74]. We therefore did not include this condition in the present set of experiments.

## Alternative explanations for omission responses

Neural circuits can display dynamics extended over multiple seconds, and therefore the neural response to a stimulus - in particular in a sequence - will often be more complex than merely a response-delay-shifted scaling of the stimulus. Therefore, we used multiple control conditions to assess whether the presently observed activity qualifies as genuine omission responses, in contrast to responses generally considered more basic, such as offset responses, release from adaptive suppression, or entrainment.

Offset responses have long been known to be produced by cortical neurons at the termination of a sound (e.g., ref. 75), and similar responses are already present on the brainstem level[76–78]. The relation of offset responses to other responses has been shown to be nontrivial[51]. They are themselves interesting in the context of predictive processing, as they signal a transition into silence, and can thus be interpreted as signaling a (locally unexpected) change in the stimulus, in particular in conjunction with the typically observed adaptation of an onset response to an adapted response level during continued stimulation. The existence of offset responses necessitates the verification that the neural activity during the omission period is not just an offset response. We have substantially reduced the likelihood of this possibility, by comparing against the activity in expected omissions (Fig. 5), contrasting the spatial representation between offset and omission responses (Fig. S3), demonstrating that the omission response lasts longer and has a different shape than the offset response (Fig. S10) and that a second omission gives another elevation of activity, which is entirely unexpected in the case of an offset response (Fig. 5). The only remaining way in which the omission response could still be an offset response, would be if - through exposure to the sequence - the offset response transforms in its shape and duration into the observed omission response. Previous research

has at least suggested that local networks contribute to the creation of the offset response, which could therefore also include complex shaping[79]. While not impossible, we consider this rather implausible, in particular due to the positive inflection at the start of the omission period, as well as at the double omission control. If all of these were the case, the offset response would be so complex and adaptable that it would itself be an even more interesting, prediction-related activity.

Neural entrainment refers to the synchronization of brainwaves or neuronal activity to the rhythm of an external stimulus, such as sound[57], light[80], or tactile input[81]. Entrainment has been shown to support faster reactions and higher gain for behaviorally relevant events, and has therefore also been considered a prediction-related neural mechanism[56]. Whether entrainment is generated locally or by across-area interactions is unknown. Local sources could include frequency-dependent synaptic filtering[82]. Research in humans has suggested that entrainment does not occur in any neural population, showing that it was absent in the brainstem, while present in the auditory cortex under the same paradigm[71]. The hallmark of entrainment is that the response to the repeated stimulus becomes stronger and/or better aligned in phase, possibly due to an induced oscillation of the neural population[83,84]. As this oscillation could continue during an omission, the OR could be a consequence of entrainment. We analyzed whether entrainment occurs in our data, finding that this is possibly the case in the ORR (Fig. S9D–F); however, the omission response does not resemble the entrained stimulus response (Figs. 1, S9G, H): while the stimulus response is a fast, bipolar signal, the omission response is a slow, unipolar signal. We find it noteworthy, though, that at the end of the sequence, the final OR shows suggestively spaced peaks (although not exactly at the expected stimulus times). However, given the low number of repetitions, these could equally likely be random fluctuations.

Related to this observation, the echo responses mentioned above appear to be similar to the presently reported omission responses, i.e., they are quite salient and have a specific temporal onset aligned with the expected time of the next stimulus. However, echo responses have only been demonstrated for quite slow (0.5 Hz) stimulus sequences[28,29], which were not tested here, as intermediate (SOA 400) sequence speeds already showed substantially reduced omission responses[25]. Additional studies that run the corresponding paradigms in the same prep and record from the same cortical regions are required to clarify their relationship.

In summary, we consider both offset and post-stimulus entrained neural responses to be distinct from the presently observed omission responses, based on their shape, timing, and evolution over the sequence. However, these may potentially be more basic variants of a general, possibly prediction-related processing principle, and therefore not clearly distinguishable from omission responses in general (see also below).

### Implications for predictive processing theories of neural function

Predictions - formed from recent sensory input or long-term statistics - are an important factor for understanding human and animal behavior[4–6,8,10,13,21], and see the "Introduction" section. Correspondingly, the underlying neural representation has been of high interest to neuroscience over the past decades. Probably the most widely cited theory has been predictive coding[2,5], and the related free energy principle[3]. While in their original formulation, non-dynamic, recent developments have extended these predictive frameworks to dynamic context, e.g., the PredNet model[85] or Dynamic Predictive Coding[86]. In parallel, over the past years, Temporal Prediction has been proposed as an alternative organizing principle which assumes that receptive field properties of neurons are optimized to present the predictive features of sensory input[60,61]. Further, neuroscientists have extended and adapted these theories to account for their experimental observations[7,26,87,88]. Together, these theories make various predictions in the specific context of the present experiment, which allows the present data to provide supportive or contradictory evidence for the above theories. Specifically, we will focus on the observed temporal shape and polarity of the omission response and its cellular implementation.

In the classical account of Rao and Ballard[2], the prediction about the stimulus could be represented as an *elevation* of a neuron's firing rate. Generalizing this to time-varying predictions, we would expect this elevation in firing rate to exhibit a time-course that matches the time-dependent occurrence probability of the sensory stimulus. This prediction, if subtracted - e.g., via an inhibitory interneuron/connection - from the sensory representation in another neuron, would ideally completely extinguish the time-dependent stimulus response, leaving no prediction error, leading to high redundancy reduction[89]. This kind of computation appears to be realized in the dopaminergic neurons in the ventral tegmental area[90], whose activity can be interpreted as the difference between expected and received reward, with the expectation being represented by local inhibitory neurons[91].

In the present setting, predictive coding would suggest a representation of predictions or prediction errors in different neurons: (i) Predictions would ideally be observable as elevated firing rate, time-limited for the stimulus duration (75 ms); while (ii) Prediction errors would be observable as a time-limited *reduction* in firing rate, as the prediction is subtracted from the spontaneous activity in silence.

While the OR shows a timed onset, it substantially outlasts the stimulus duration. In terms of direction, it is positive, and therefore only compatible with predictions, or a separate representation of positive and negative prediction errors, as proposed previously[26,88]. If the presented data only came from neurons representing negative prediction errors, this would raise the question of where the neurons representing predictions and positive prediction errors are, given that the vast majority of cells in the ORR show an elevation of the response (Fig. 2, and also ref. 26). Further, the sign inversion for subtracting the prediction would likely be realized by inhibitory neurons, which constitute only ~20%[92].

An alternative to a network architecture underlying predictive coding has recently been proposed by Mikulasch et al.[7], who suggested that the combination of top-down predictions and bottom-up input could be performed via apical and basal dendrites, respectively. In this model, prediction errors remain confined to subthreshold potentials, while predictions materialize as action potentials. While this would be consistent with the present large majority of cells showing an OR, it would still not account for the OR's time-course.

Finally, another possibility would be that prediction errors are, in fact, represented; however, only in terms of their absolute value. Further, the ramping shape of the OR, including the additional increase at the second omission (Fig. 5G–K), suggests an integrated signal, which would thus be consistent with a summed, absolute mismatch between the prediction and the input. To our knowledge, this interpretation has not yet been explored in models of predictive processing; however, it appears to be most consistent with the present results. Interestingly, the recent concept of Temporal Prediction[60,61] might provide a substrate for learning omission responses with compatible time courses[93], Extended Data Fig. 2, shown there for the case of the retina; however, additional research is required to clarify its precise origin and interpretation.

### Limitations and future directions

Facial imaging demonstrated that the mice represent the recent sequence statistics indicated by a slow, but clearly omission-associated arousal response (pupil dilation) as well as a history-dependent, stimulus-driven facial motion response. The latter increased after omissions and scaled with longer sound sequences before the omission. While challenging in the present, rapid sequence design, integrating an active variant in the experimental design that requires the

mice to respond to omissions would ensure that they are attending to the acoustic sequence, which often leads to enhanced responses[94] and could provide more clarity on the relationship of the neural and behavioral correlates of omission responses and enable distinction of prediction monitoring from arousal responses. Likely, a modification to a different stimulus design would be necessary to allow an animal to respond to omissions, without the response itself influencing the quality of the neural data.

We chose the recent jGCaMP8m calcium indicator due to its high signal-to-noise ratio and fast onset and offset dynamics, which are essential for the present study. An alternative would have been the even faster variance jGCaMP8f, which, however, is ~50% less bright than jGCaMP8m, which would come even closer to the temporal precision of electrophysiology. While we have verified that at the current 30 fps in 2p imaging, deconvolution does not improve the quality of the readout, imaging selected cells at higher rates would enable single spike deconvolution[35], and could provide temporal accuracy in the low millisecond range.

We used viral transfection for expressing the indicator in the AC - leading to partly inhomogeneous expression and coverage across animals - as no normal-hearing, transgenic lines were available to us at the start of our study. In future studies, it would be preferable to make use of retroorbital PHP.eB vectors[95], neonatal intraventricular injections[96] or tail vein injections[97], or directly switch to transgenic lines[36,98], in particular the more recent soma-targeted RiboL1 variant of jGCaMP8m[95]. In addition to more homogeneous expression, this would improve the SNR in 2p imaging, likely allowing larger fields of view and thus larger populations to be imaged. While these methods come with their own caveats, they share the advantage that imaging can start directly after the window implant surgery.

The presently used transfection approach labels all neurons irrespective of their type. In relation to theories of predictive processing, obtaining separate results for different cell types would be highly important, as inhibitory interneurons would be expected to play a specific role in the combination of (top-down) predictions and (bottom-up) sensory information[88,99,100]. As recently shown by Tsukano et al.[101], even seemingly simple, adaptive processes can be primarily non-local, in this case realized by a top-down circuit from the orbitofrontal cortex that specifically targets SST cells in the AC. While C57Bl/6-based transgenic mouse lines are available now[98], these develop age-dependent hearing loss, which makes them a suboptimal model in auditory neuroscience, leaving Cre-Lox-based transfection as an alternative (see above for achieving homogeneous expression). Further, our results indicate that further insight could be gained by expressing the indicator in specific cortical layers, cell types, or optogenetically manipulating interareal projections[102].

## Methods

### Experimental model and subject details
All experiments were performed at the Central Animal Laboratory at the Radboud University Medical Center. The experimental procedures and protocols were approved by the Dutch central commission for animal research (Centrale Commissie Dierproeven, project numbers: 2017-0041, 2023-0031) and implemented in accordance with the local animal welfare body. Female adult CBA/JRj mice (Janvier Labs) were co-housed in a normal 12 h light/dark cycle and provided with *ad libitum* food and water. CBA/JRj mice display normal hearing for many months, in contrast to C57Bl/6 derived strains, which develop age-related hearing loss at high frequencies early on[36,103]. Surgeries were performed when the mice were 6-7 weeks of age, and all recordings in this study were obtained within 10 weeks after surgery.

### Implant and transfection surgery
Mice received pain medication via the drinking water (Carprofen, 5 mg/kg, based on regular water intake) in the 24 h before the surgery.

At the start of the surgery, mice were anesthetized with isoflurane (3% induction, 1–2% for maintenance) in equal parts oxygen and air. Breath rate was monitored to range between 40 and 80 bpm. Carprofen (5 mg/kg) and Dexamethasone (2 mg/kg) were administered i.p. after induction of anesthesia. Core body temperature was maintained using a homeothermic blanket (RWD RS-485) in combination with a rectal thermometer, and 0.5 ml saline injections were administered s.c. per hour of anesthesia.

Scalp fur was removed using a depilatory cream, and the skin on the scalp was disinfected with Betadine and ethanol (70%). Local anesthesia was provided by injecting a mixture of Lidocaine/Bupivacaine (0.1 ml) s.c. using a fine syringe needle. A surgical incision was made over the midline. The right hemispheric portion of the skull was first exposed, and the periosteum was removed using a spudger in combination with 3% $H_2O_2$, after which the skull was slightly scratched using a scalpel. A custom-made aluminum headpost was implanted on the right skull hemisphere at a ~45° angle away from the medial axis using dental cement (Superbond, Sun Medical). Next, the left skull hemisphere was exposed down to the ventral portion of the temporal bone and cleaned with 3% $H_2O_2$.

The craniotomy center target was 2.4 mm posterior and 4.7–5.0 mm lateral, about 0.5 mm medial to the parietal-temporal ridge, measured point-to-point relative to Bregma using a crosshair laser (650 nm, 5 mW, Quiaoba) mounted perpendicularly to the skull top on a micropositioner (Kopf Instruments). Prior to drilling, the skull around the craniotomy was cleaned, scratched, and stabilized using dental cement. Depending on the cranial window used, a craniotomy with ø = 3–4.3 mm was performed around the estimated center using a combination of a micromotor drill (K.1070-2, Foredom) with a 0.5 mm steel burr (19007-05, Fine Science Tools) and a #4 forceps (Fine Science Tools). To minimize dura and vessel trauma, the central bone piece was lifted out using a handle cemented onto the piece's center. Once exposed, the dura mater was kept moist by regular application of sterile saline droplets.

For transfection with the activity reporter jGCaMP8m, a pulled glass micropipette (5 µl capillary, inner tip diameter ~20 µm, unbevelled, 555/5, Assistent) was back-filled with mineral oil and then loaded with ~2.5 µl viral solution ($2 \times 10^{12}$ vg/ml; pGP-AAV9-syn-jGCaMP8m-WPRE; #162375, Addgene). Using a micromanipulator (SMX series, Sensapex, Finland), the micropipette was angled at ~45° with respect to the brain surface, and injection locations were offset medially to result in an 800–1000 µm spaced honeycomb pattern centered on the craniotomy at ~400 µm cortical depth (800 µm injection depth). Per site, 150–250 nl viral solution was injected at a speed of ~30–60 nl/min, followed by 4 min pauses before pipette retraction, to avoid backspilling.

A cranial window was constructed by gluing a stack of 2 small (3/3.5/4.3 mm) and 1 larger (4/4/5 mm) glass platelets (#0 Thickness, Warner instruments), using optical adhesive (NOA68, Norland), and stored in 70% ethanol. It was rinsed with double-deionized water and then cemented into the craniotomy, gently depressing the tissue. Finally, a custom-made, grade 5 titanium headplate was cemented planar with the window and in alignment with the skull's central fissure using the micropositioner, to ensure accurate positioning across multiple imaging sessions. See S4 for a visual summary of the surgical procedure.

The animals were allowed to recover from the surgery for a week and were then gradually habituated to head fixation and the recording setup for at least one more week, using sugar water as a reward. Viral expression levels were monitored from the second week onwards until they stabilized, before commencing the experiments.

### Widefield imaging
Widefield recordings were performed in a dark and sound-insulated chamber, using acoustic foam on the inside (thickness: 5 cm, black

surface, Basotect Plan50, BASF). The acoustic foam shields against external noises above ~1 kHz with a sound absorption coefficient >0.95, defined as the ratio between absorbed and incident sound intensity. This corresponds to >26 dB of shielding in addition to the acoustic shielding provided by the booth itself. A 470 nm LED (M470L4, Thorlabs) was triggered at 100 mA for intervals of 1 ms at 100 Hz, using a multichannel LED driver (DC4140, DC4104, Thorlabs). The light was passed through a collimation lens (ACL2520U, Thorlabs) and focused on the brain surface vessels using a 4× objective (Plan Fluor 4×/0.13, MRH00041, Nikon), resulting in an illumination intensity of 0.378 mW/mm². The emitted light was filtered for its green light components using a longpass dichroic mirror (DMLP490R, Thorlabs) and a bandpass filter (MF525-39, Thorlabs) and recorded using a high-speed, global shutter sCMOS camera (U3-3060CP-M-GL, IDS Imaging), recording at 100 fps at 1 ms exposure duration and a resolution of 1920 × 1200 pixels, spatially averaged 4× in both dimensions during acquisition. To account for idiosyncratic differences in fluorescence levels between animals, the camera's digital gain factor was adjusted between 1 and 3 to produce a raw signal intensity with maximum pixel intensity at 25–50% of the sensor's bitrange. Overall photobleaching of the fluorophore intensity was very slow and amounted to <5% of the overall signal intensity over a whole recording session (2–3 h), and was therefore deemed insignificant over the much shorter period of single paradigms. Imaging data was time-aligned per frame to the clock of a multichannel data acquisition device (PCIe-6351, National Instruments) at 10 kHz.

## Two-photon imaging

We performed cellular resolution, 2-photon imaging in the same animals, guided by the response maps obtained in widefield imaging. An infrared laser (Chameleon Vision-S, 925 nm, 1467 mW, Coherent) was pulsed at 80 MHz for 75 fs intervals, attenuated with a Pockels cell to 7-90 (mean ~30) mW in the focal plane, depending on the imaging depth and indicator expression levels. The laser's group delay dispersion was adjusted for maximum fluorescence signal to compensate for chromatic aberrations in the light path. A sound-shielded (5 cm acoustic foam, Basotect BASF, inside a wooden box), resonant-galvo scanner (6SD11798, Novanta) in combination with a photomultiplier tube module (H11706-40, Hamamatsu) was used to image at 30 Hz (dwell time: 127 ns/pixel) at a resolution of 512*512 pixels, covering an area of 485×490μm. The imaging path was calibrated such that the back objective (16× LWD, 0.8 NA, Nikon) aperture was ~120% filled with the expanded laser beam, and the half-width of the point spread function was <8 μm in x and y and <15 μm in z for a 4 μm fluorescent microbead. As the animals were held at their headpost in a near upright position, the scope was tilted 38° off the vertical axis. The animals were head-fixed on a 3D-printed bed and restrained in order to minimize the chance of harm to the animal or motion artifacts. A goniometer was added below in order to allow for fine-tuning of the mediolateral tilt. Centrifuged ultrasound imaging gel was used as an immersion medium, and a custom 3D-printed matte-black sleeve was placed on the objective to bridge the gap to the sample and protect from residual stray light. To minimize shot noise and auditory noise, lights were turned off in the room during all recordings. A laser-safety tent as well as a sound-insulating enclosure (5 cm acoustic foam) around the animal were used to protect the objective from external light and sound sources. Further, the platform motors (X-LRM050A-E03-ENG2333 and X-LRQ050AP-DE51-ENG3237, Zaber Technologies) were sound-shielded with acoustic foam. Acoustic background level was verified to be rather flat inside the enclosure using an ultrasonic microphone (see below), and in particular did not contain a peak at the resonant frequency of the scope (8 kHz). All data acquisition was controlled through the ScanImage (Premium, 2023.1.0, MDF Bioscience[104]) software system inside Matlab. Imaging data was time-aligned per frame to the clock of a multichannel data acquisition device (USB-6251, National Instruments) at 10 kHz.

## Pupil and facial imaging

During widefield recordings, we recorded from the right snout and eye simultaneously, i.e., opposite to the craniotomy to avoid mixing of light sources. Images were acquired using a high-speed camera (GV-5240CP-NIR-GL, IDS, Germany) equipped with a camera zoom lens (XM-300, SainSonic) under infrared illumination (850 nm) using an LED (M850L3, Thorlabs) powered by an LED driver (LEDD1B, Thorlabs) collimated by a lens (ACL2520U, Thorlabs). Images were captured at 50 fps and an exposure duration of 15 ms. Precise temporal alignment was performed by collecting the frame triggers at 10 kHz in the common DAQ device (see above). After the recording, images were compressed to MPEG-4 with the constant rate factor set to 20 and stored for later processing.

## Acoustic stimulation

Sound stimuli were generated and recorded using a multichannel data acquisition device (USB-6251, National Instruments) through a custom-written software package (*Controller*, implemented in Matlab R2018b, The Mathworks). Sounds were presented to the right ear using a high-quality, miniature driver (IE800, Sennheiser) placed at a 2 cm distance from the pinna at a sampling rate of 250 kHz. The output of the speaker was calibrated to have a flat spectral profile over the range of 2–70 kHz within 5 dB using an inverse linear filter (see ref. 105 for details), estimated before the experiment using an ultrasonic free-field microphone (Type 4939, Bruel & Kjaer), amplified by a signal amplifier (Model 1708, Bruel & Kjaer) corrected for its own frequency-dependent sensitivity curve.

## Acoustic paradigms

The sounds used for stimulation were of 75 ms duration, including sinusoidal ramps of 5 ms at their beginning and end, presented at 50 dB SPL. The sinusoidal tone frequency was either 4 kHz or 32 kHz for the pure tone paradigms, and white noise stimuli were presented with constant power spectral density between 2 and 64 kHz.

For the oddball paradigms, tones were presented in regular intervals with a stimulus onset asynchrony (SOA) of 100/200/300/400 ms, i.e., the time between subsequent stimulus onsets. For the SOA Jitter control paradigm, SOA was pseudo-randomly jittered around 200 ms by ±[0,50,100] ms, such that each individual SOA from the set {100,150,200,250,300} was presented equally often. An oddball was either a silence (in the case of stimulus omissions, i.e., amplitude set to 0, equal to -Inf dB) or a non-standard pure tone (i.e., 32 kHz or 4 kHz in the Deviant paradigms, depending on the frequency of the standard). The presently used separation between deviant and standard (3 octaves) is larger than typically used in single-cell studies (<=0.5 octaves[16,106–108]). We chose a large separation to be able to study large parts of AC in the same paradigm; however, this means that the deviant and standard will typically activate different neural populations. We therefore refer to the deviant as a *distant deviant*, to avoid confusion with the classical use of deviant in the literature, which refers to a *local deviant* that is within the tuning curve of cells that also respond to the standard.

After a lead-in sequence of 20 repetitions of the standard stimulus, oddball stimuli were interjected after a pseudo-randomly chosen number of standards. In each trial, the number of standards between oddballs was equally distributed over the integers [1,2,3,4,5,7,9,11,13,15], such that each number of standards appeared 10 times, resulting in 100 oddball presentations per trial, reduced to 81 after excluding sequences with at least 2 standard tones before the oddball. In a subset of animals, we also recorded additional paradigms which included double omissions, i.e., 0 was included in the above set. Trials were thus rather long sequences (820 stimuli, 82–328 s for 100–400 ms SOA, respectively), with the purpose of establishing and maintaining the regularity of the standard sequence. Per parameter set, two trials were run. Paradigms were only directly compared if they were recorded in the same imaging session.

For the Many Standards Control sequences, the probe tone was presented in place of the oddballs, and for each standard, tone frequency was pseudo-randomly drawn from 10 frequencies, log-spaced between 2 and 64 kHz (excluding the probe frequency), such that each frequency appeared with equal likelihood and perceptually at random.

For the tonotopy estimates, 75 ms pure tones with 11 log-spaced frequencies between 2 kHz and 64 kHz were presented at 4 different amplitude levels, linearly spaced between 20 and 80 dB SPL. The SOA was 1 s, though we verified that SOAs of 500 ms and 50 ms produced qualitatively similar results. Per trial, each frequency-amplitude combination was repeated 6 times in differently randomized order, with each randomization repeated 5 times. This design was chosen to allow for post-hoc exclusion of both sequence-specific and unscrambling-related effects. Across trials, each frequency-amplitude combination was presented 30 times.

## Data analysis

*Widefield image processing:* Frame times were synchronized to stimuli based on the NI frame clock timestamps. For rare cases in which a frame was lost during acquisition, the missing frame was replaced by the previous frame to avoid using future data. Images were down-sampled by a factor of 2 to a final resolution of $240 \times 150$ (MATLAB: *imresize*). Per trial, the fluorescence baseline of each pixel ($F_0$) was estimated with an 8 s moving median (symmetrically around each time step) and used to calculate the pixel's relative luminosity over time ($\Delta F/F_0$). We verified that using a single pretrial $F_0$ yielded qualitatively matched results. The moving median baseline provided lower variability in comparison and was therefore chosen. Comparing baselines estimated between trials and animals, we found the baseline to vary over time and between trials, meandering both in the positive and negative direction, both under stimulus-driven and silent conditions (for details see Fig. S7). On average, only a small, systematic decay was observed over the length of the trial (~150–330 s), which we attribute to a slight reduction in indicator brightness - possibly bleaching - over time. In this study, no spatial or temporal deconvolution was performed in order to prevent assumptions on the spatial light mixing or the indicator dynamics, respectively, to bias the readouts of spatio-temporal activity profiles. No temporal low-pass filtering was performed as the onset dynamics of jGCaMP8m provided informative signals up to the full sampling rate of the camera (100 fps). To verify time alignment and sequence unscrambling, we simulated video data based on the raw stimulus waveforms for Oddball and Tuning paradigms and applied our analysis to these. To minimize the effect of small motion artefacts (typically 1 pixel or less) on dF-derived functional metrics on vessel outlines, sub-pixel rigid motion correction (NoRMCorrE package[109]) was used.

*Two-photon image processing:* Preprocessing and ROI identification and extraction were performed in *suite2p* (V0.12.0[110]). Specifically, we performed the following steps:

(i) Motion correction via nonrigid frame registration. Parameter settings: `nonrigid=1`, `block_size=[128,128]`, `snr_thresh=1.2, maxregshiftNR=5`.

(ii) Functional cell detection. Parameter settings: `roidetect=1`, `denoise=1`, `spatial_scale=0`, `threshold_scaling=1.0`, `max_overlap=0.75`, `max_iteration=20`, `high_pass=100`, `spatial_hp_detect=25`.

(iii) ROI selection: Due to variable image quality, manual sub-selection was performed according to the mean image after the functional cell detection. This avoided the inclusion of ROIs with poor quality spatial profiles or locations, i.e., non-cell-like shapes, overexpressed cells, or cells located inside vessel shadows.

(iv) Cell signal and neuropil extraction. Parameter settings: `neuropil_extract=1`, `allow_overlap=0`, `inner_neuropil_radius=2`, `min_neuropil_pixels=350`. Neuropil

decontamination is performed by selecting a correction coefficient β according to the correlation between the extracted raw cell signal F and the corresponding extracted neuropil signal $F_{neu}$. The decontaminated signal was computed as: $F_{dec} = F - \beta \cdot F_{neu}$, where F is the raw cell trace, and $F_{neu}$ is the corresponding neuropil trace. The coefficient β was estimated from a linear regression of F on $F_{neu}$: $F \approx \alpha + \beta \cdot F_{neu}$. Since calcium signals can be considered mean-centered, the intercept α was omitted. Under this assumption, the least-squares estimate of β is: $\beta = Cov(F, F_{neu}) / Var(F_{neu})$, which can also be expressed as: $\beta = r \cdot (\sigma_F / \sigma_{Fneu})$, where r is the Pearson correlation coefficient and σ denotes the standard deviation of each signal. In our dataset, the correlation between raw and neuropil traces was relatively low ($r = 0.2 \pm 0.1$), resulting in β values typically in the range of 0.3–0.5.

The cells' spatial profiles, locations, and corrected activities were then collected for further processing. No further temporal filtering or deconvolution was applied to the $Ca^{2+}$ signals to analyze the temporal information in the signal without distortions. To avoid biases due to decreasing single-cell quality with depth, we also analyzed the 2p data using a grid-based approach, specifically, each of the raw 2-photon image frames was evenly divided into a $16 \times 16$ grid, each block containing $32 \times 32$ pixels of the original image. Within each block, the combined signal was computed as the mean value of all pixels within the block of the frame.

Traces extracted were then aligned to the central system clock by recording the frame timestamps on the central DAQ device. Sub-sample alignment correction was performed for different locations inside an FOV to account for the sequence of scanning by the laser, which started in the top-left corner and progressed to the bottom using bidirectional scanning. In total, this amounted to a maximally corrected delay of ~30 ms (-1/frame rate), which is relevant given the rapid onset dynamics of $[Ca^{2+}]_{in}$ as well as jGCaMP8m (<5 ms[35]). For display purposes, data from different neurons were then commonly binned, and data displayed at a given time correspond to the last scanning time-point before this time. This ensured that no data from the future contributed to the data displayed at a given point in time (which would have been the case when using binning based on minimal distances to bin centers).

Deconvolution was deemed not necessary due to the fast time-scales of the indicator (2.5 ms @ 50% rise time, and ~40 ms @ 50% decay time[35], their Extended Data Table 6) in relation to the 2p imaging time resolution (33 ms). We have verified that deconvolution did not change the results qualitatively, in particular, not the omission responses.

## Spatial alignment between widefield and 2p data

For each mouse, a high-resolution vessel image of the craniotomy was used to determine an individual unique vessel landmark. At the start of each two-photon imaging session, the center of the imaging field was aligned with this landmark in x, y, and z using widefield illumination, and the scope's motor coordinates were recorded. With the relative coordinates computed by subtracting landmark coordinates from the current coordinates and taking into account the relative rotation and scaling caused by the scope's tilt (38°) and the zoom factor (set via the range of the scan mirrors) respectively, all cells' location including their profile in subsequent imaging fields can be mapped onto a coordinate of the craniotomy at the accuracy of μm.

## Tonotopy estimate

To sharpen the pure tone response estimate, the mean over a period of 200 ms preceding stimulus onset was subtracted from each pixel's $\Delta F/F_0$ trace per frequency-amplitude combination per trial. Responses were collected over a time window of [0,230] ms, relative to stimulus onset (100 ms stimulus duration plus a 130 ms) to account for the

dynamics of response and calcium indicator, based on rise-to-peak and half-decay constants reported in ref. 35, over which the maximum value was taken as the response. After taking the mean response across trials, frequency-amplitude combinations were excluded if their mean value did not exceed the 55th percentile of the pixel's responses across frequencies and amplitudes. The best frequency was then computed using the approach in ref. 36: The first amplitude with an above-threshold response was defined as the hearing threshold for that pixel, and responses were summed over that and the two higher amplitudes, excluding amplitudes above 70 dB SPL. Across frequencies, the weighted normalized average was then taken per pixel, and the maximum was used to select the best frequency. The degree of responsivity per pixel (shown in Fig. 1D as alpha scaling) was computed as the 55th percentile of responses across amplitudes and frequencies. While tonotopies exist and can be recovered, tonotopic maps vary substantially between animals and do not directly match the temporal organization of the auditory cortex and its surrounding areas, consistent with the variability in auditory cortex location and internal division described previously[36,111].

### Region definitions

As there is no widely accepted consensus as to the precise defining criteria for the primary and secondary auditory fields[36,38,40,112], we developed an approach that combines conventional area estimates through tonotopy and low-frequency response centers with an estimate for temporal fidelity, since secondary cortices by definition exhibit slower response dynamics[38]. We quantified this by measuring the peak-to-trough distance (in $\Delta F/F_0$) that a pixel exhibits during a single noise burst presentation (100 ms, 50 dB SPL), as well as the half-max latency using a Fourier transform interpolation of the trace (MATLAB interpft) surrounding the response peak. These measures resulted in stable and well-defined localization of core, fast-responding response areas (encompassing AI, AII, and AAF) due to the narrow time window of the stimulus and their fast rise-to-peak dynamics. Since GCaMP6-based imaging studies do not provide the opportunity to resolve these temporal response characteristics, we argue that previous estimates of the spatial extent of primary auditory cortices may have been overestimated, and we instead characterized some of the posterior tonotopically organized areas as belonging instead to TeA based on their weak tonotopy, long response latency and modulation speed, as well as weak impulse responses, in accordance with classical anatomical labeling in the Allen Brain atlas[113], as well as a recent study which characterized TeA with sparse, high-latency auditory responses[37]. Using a custom-written graphical user interface (GUI), we defined area borders by hand per animal. The approach was to define the AI-AAF intersection in the high-fidelity center, and then constrain further border definitions to a 1–1.5 mm diameter and fidelity of the primary regions, using tonotopical features as augmentations to guide further area border choices. The extent of the secondary cortices was estimated by using the normalized global (i.e., post-stimulus onset) maximum response value to a 100 ms noise burst presented at 50 dB SPL and empirically setting the threshold to the 75th percentile of maximum values outside of the craniotomy (see Supplementary Fig. 4 for an example of the region selection for the animal shown in Fig. 1). This metric for responsivity was then normalized and smoothed with a 3-pixel Gaussian kernel (MATLAB imgaussfilt) to generate an alpha map (see Fig. 1E).

### Pupil tracking and facial motion analysis

The videos obtained from facial imaging were used to quantify pupil size and facial motion. Pupil size was estimated by tracking multiple points on the pupil perimeter and the eye using the SLEAP software package, a deep learning framework that enables animal pose estimation[114]. Specifically, eight nodes on the pupil edge were tracked to cover the horizontal, vertical, and diagonal distances. Furthermore,

there were four nodes placed on the left, right, top, and bottom of the eye to establish when the mouse blinked. The model that was used to track these nodes throughout the videos was trained on 426 labeled instances across different animals and recordings. The final pupil size was defined as the median Euclidean distance between all opposite points on the pupil edge. Frames in which the eyelid was partially closed and the pupil thus (partially) occluded were excluded and interpolated between neighboring frames.

To extract a scalar measure of facial motion, we quantified the overall amount of change between subsequent frames[115]. Specifically, first frame brightness was standardized to the interval [0,1] per recording, by using a random subset of frames to discretize the intensity into 100 bins, and the remaining frames were then mapped to these bins. Next, the motion energy per pixel was computed as the absolute, pixel-wise difference between subsequent frames. This measure was normalized per recording by subtracting the pixel-wise mean over time. Finally, overall facial motion energy (FME) was quantified as the mean across pixels to obtain a scalar measure over time. We also performed SVD-based (singular value decomposition) decomposition into facial motion components[49], however, this provided qualitatively similar results and was thus not included in the results.

### Oddball response extraction

For the presentation of each standard stimulus or oddball - tone or omission - the corresponding frame times were selected, including frames that fell into a pre- and post-period of 3 times the SOA. For averaging, only repetitions of the standard or oddball were included that had ≥2 standard stimuli preceding and following them. This ensured that the average only contained averages at each point in time over (locally) identical stimulus conditions. The corresponding extraction of frame times was performed for the different data sources alike, i.e., widefield, 2p, and facial video, used for extracting pupil and facial motion data. The average omission response (OR) was calculated as the mean over the response values from the onset of the omission to the onset of the post-omission stimulus. The omission-responsive region (ORR) was determined using the OR. Specifically, we calculated the upper 98th percentile of the omission response values. To refine this region, we applied morphological operations, including dilation and erosion, and Gaussian filtering to smooth the boundaries and remove gaps within the mask.

### Spatial clustering of neural responses to double omission

Widefield imaging data was acquired as a three-dimensional array with spatial dimensions $X \times Y$ and temporal dimension $T$. To identify spatially distinct functional regions related to the double omission (dOM) paradigm, k-means clustering was applied to the time activity profiles of each pixel. Specifically, the temporal responses during the dOM paradigm were used to cluster pixels into an empirically selected number of groups ($k = 9$) based on the similarity of their activity patterns. Each pixel was assigned to one of the nine clusters according to its temporal response characteristics. A cluster assignment map was then generated by color-coding each pixel according to its cluster label, producing a spatial representation of functionally distinct regions during the dOM paradigm.

### Entrainment analysis

We defined a custom score to quantify entrainment. $\Delta F/F$ traces were aligned to stimulus onset. For each animal, upper and lower envelopes were obtained by interpolating between the identified largest local peaks and troughs of the $\Delta F/F$ trace in response to each stimulus. The average of the upper and lower envelopes was defined as the overall response component (Fig. S9C), i.e., ignoring the rapid responses to each stimulus. This component was subtracted from the original trace to remove overall response effects prior to calculating entrainment

scores. The entrainment score for each stimulus was defined as the difference between the peak and trough within a window extending from stimulus onset to 50 ms after stimulus offset (truncated to 25 ms for 100 ms SOA sequences). Per-animal scores were normalized by dividing raw values by the median entrainment score of the primary region across all stimuli. For omission analysis (Supplementary Fig. 9G), entrainment scores were calculated in the same way as for stimulus periods, then averaged across all omission events within each animal before computing the across-animal mean. In all other analyses, scores were averaged directly across animals after per-animal normalization. Differences between conditions (stimulus vs. omission, PR vs. ORR, or SOA dependency) were tested using linear mixed-effects models (fitlme, MATLAB) with condition as a fixed effect and animal as a random intercept; p-values for fixed effects were taken from the ANOVA table of the model.

### Statistical analysis
We performed mostly non-parametric statistical comparisons to avoid distributional assumptions, which are hard to assess on the animal level ($n = 10$). Therefore, we mainly used the Wilcoxon Rank Sum and Signed-Rank test for unpaired and paired comparisons, respectively. Further, for multigroup comparisons, we first used the Kruskal–Wallis ANOVA, using Bonferroni correction for multiple testing. Test outcomes were considered to be significantly unlikely given the null-hypothesis if the p-value was below 0.05 after correction for multiple testing, which is indicated with an asterisk in figures. Two asterisks indicate a p-value less than 0.01, and three indicate a p-value less than 0.001. If different testing strategies were required, this is specifically indicated in the text. Error bars/error hulls show standard errors of the mean, i.e., standard deviation divided by the square root of the number of animals/cells, unless indicated otherwise.

In comparisons involving cells from multiple animals, we used a nested testing approach, i.e., linear mixed-effects models (fii.tlme, MATLAB), with one fixed effect (e.g., region) and animal as a random intercept. P-values for the fixed effect were obtained from the ANOVA table of the mixed model. We checked the degree of normality of the data (Shapiro–Wilk test combined with the Quantile-Quantile plot), which was typically the case. Due to the lack of a non-parametric alternative for fitlme in MATLAB and the high degree of significance of the comparisons, we applied the assumption of normality in all cases.

### Subject exclusion criteria
In total, we performed implant surgeries and ran all/subsets of the above paradigms in 14 mice, out of which 4 were excluded from further analysis if either the auditory cortex was not widely enough transfected (1), there was tissue infection and/or dural regrowth prohibiting imaging (2), or due to implant failure (1).

### Histology
Based on earlier research[116], AAV9 and AAV1 were candidate vectors for transfecting cortical neurons with the jGCaMP8m construct. In order to verify functionality and relative strength of the vectors, slices of cultured rat hippocampal neurons were transfected using jGCaMP8m in AAV1 and AAV9 serotypes (Addgene #162375, 1×10^12 vg/ml). Both vectors appeared to work equally well in vitro (see Supplementary Fig. 5A/B), and the AAV9 construct was chosen for further testing. To verify that the construct expresses as intended in vivo, one animal was transcardially perfused with DBPS and PFA after termination of the in vivo experiment, and brain slices from -3.2 mm posterior to bregma were stained using rabbit anti-GFP and anti-rabbit GFP to enhance the signal-to-noise ratio of jGCaMP8m's intrinsic fluorescence in DPBS[34]. Slices and fixed culture preparations were imaged using a widefield microscope (Thunder, Leica), to confirm that viral expression is localized to all cellular compartments of cortical neurons in the mouse after in vivo injection (see S5C).

### Reporting summary
Further information on research design is available in the Nature Portfolio Reporting Summary linked to this article.

### Data availability
The neural and video data generated in this study, as well as all files necessary for the creation of all figures, have been deposited in the open-access Radboud Repository (doi: 10.34973/e6ph-q277) in a compressed (i.e., preprocessed) format. Source Data for each figure is organized under Repository/Source Data/FigureName/[..]. For more details on data structure, refer to the README.txt file in the base directory of the repository. The raw neural and video datasets were too large for online storage, but will be made available upon request to the corresponding author.

### Code availability
All code used for stimulus generation, data acquisition, preprocessing, data analysis, and figure plotting is also made available via the same Radboud Repository as the Data, fixed from the time of publication (doi: 10.34973/e6ph-q277). Inside /Code/, we provide code for the creation of each figure separately under the subfolder /Figures/. Sub-folders inside /Figures/ are structured per Figure, according to the format 'Figure_X_ShortTitle'. For more details on code structure, refer to the README.txt file in the base directory of the repository.

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

## Acknowledgements

We would like to thank Jeroen Bos and the support staff of INSS for their assistance during the development of the experimental setup. We would like to thank Floris de Lange and Uta Noppeney for helpful discussions and feedback on earlier versions of the manuscript. We would like to thank Daniel Polley, Ross Williamson, Bruno Pichler, Yannick Goullam-Houssen, Brice Batthelier, and Rémi Proville for initial discussions on the experimental setup design and Roberta Müller for assistance with the illustrations of experimental setups. Further, we would like to thank Gesa Berretz, Karol Przewrocki, and Artoghrul Alishbayli for insightful discussions. Bernhard Englitz acknowledges funding from a VIDI grant (016.VIDI.189.052) and Zhongnan Cai from an internal grant at the Donders Center for Neuroscience. Bernhard Englitz also acknowledges valuable discussions at the Kavli Conference on Statistical Learning in the Brain at UCSB, supported by NSF Grant No. PHY-1748958 and the Gordon and Betty Moore Foundation Grant No. 2919.02 to the Kavli Institute for Theoretical Physics (KITP). Francesco P. Battaglia acknowledges funding from the European Research Council (ERC) Advanced Grant "REPLAY-DMN" (grant agreement no. 833964).

## Author contributions

Bernhard Englitz acquired funding, conceived and implemented the experiments, managed and supervised the project, and wrote the manuscript. Janek Peters developed, managed, and performed the surgeries and imaging experiments, wrote analyses and generalized analysis tools, co-developed and managed the code pipeline, supervised students, produced figures, and co-wrote the manuscript. Zhongnan Cai co-developed the analysis pipeline, in particular, all aspects of 2P data processing and visualization, and contributed significantly to manuscript writing. Maxime van Veghel developed pilot analyses for the omission control paradigms. Andreas Knoben implemented facial and pupil tracking and piloted analysis. Maikel Simon acquired imaging data. Samuel Arends assisted with facial motion tracking and pipeline improvements. Francesco P. Battaglia acquired funding for and managed the acquisition as well as the assembly of the 2p microscope. All authors read and approved the final manuscript.

## Competing interests

The authors declare no competing interests.
