## [Transparent Peer Review file · Nature Communications]

The representation of omitted sounds in the mouse auditory cortex

Corresponding Author: Professor Bernhard Englitz

Version 0:

Reviewer comments:

Reviewer #1

(Remarks to the Author)

This is a very interesting manuscript combining widefield and 2-photon imaging calcium imaging as well as pupillometry and facial movement analysis to provide novel insights into the neural representation of omitted sounds. The most significant aspect of the work is the demonstration of the localisation of omission responses to a particular region of the temporal cortex, which will be extremely valuable to those intending to dissect the circuitry underlying these omission responses.

The authors produced a solid piece of work which will certainly have an impact beyond the field auditory neuroscience. The methodology is sound and the conclusion are well supported by the evidence.

I only have some (very) minor comments.

Figure S4: put orientation bars on photos

L.59: processings > processing

Figure 1H: panel label missing

Figure 1I: incorrectly labelled as 1H

Figure S2E: I don't think I understand panel S2E and I'm wondering whether it adds much/anything to what's already shown in S2D.

Figure 2B: The ORR demarcated in 2B looks much larger than what was identified as ORR in S2D and would seem to encompass parts of A1. Or am I misinterpreting something here? I guess, to some extent, this may be explainable by differences in scale perhaps? Are all data in Figure 2 from a single animal?

L.155: more steep > steeper

L.1008: delete 'across'

L.372: an > a

L.393: times > timed

L.402: typically > typical

L.501: receptive properties > receptive field properties

L.524: suggest > suggests

L.669: used image > used to image

L.778: delete 'but'

L.785: It is somewhat unclear how the authors dealt with the neuropil contamination. Was the neuropil signal subtracted in full (F-Fneu) or was the neuropil signal scaled before subtraction (i.e. $F - 0.7 * F_{neu}$) as suggested by Chen et al., 2013 (GCaMP6) and Zhang et al., 2023 (GCaMP8). The sentence starting with "was empirically selected..." is missing at least one word at its beginning and it is not clear at all what the authors are trying to say here.

L.788: "then for further": a word seems to be missing here.

L.893: 'region' missing after 'responsive'

L.974: centrum > centre?

L.1020: I think I worked out what is displayed here (Fig 3A) but I'm not absolutely certain. This panel legend, like many others, provides the authors' interpretation of the displayed data rather than a detailed and neutral description that would allow the reader to arrive at their own interpretation. What are the axes in the top panels (there are no scale bars)? What is the green vertical line?

Figure 5J: I don't understand what is displayed here.

(Remarks on code availability)

Reviewer #2

(Remarks to the Author)

This manuscript investigates neural responses to omitted auditory stimuli using widefield and two-photon calcium imaging in mice, identifying spatially localized omission responses and proposing their relevance to predictive processing. The data are extensive and the approach is ambitious, combining neural recordings with behavioral measures (motion energy and pupil size) and a series of control manipulations. However, several technical and interpretive concerns—particularly surrounding the $\Delta F/F$ signal processing—raise questions about the robustness of some core findings. The intention here is to highlight areas where clarification or re-analysis could meaningfully strengthen the conclusions.

Calcium Signal Processing and Temporal Resolution in the Interpretation of Omission Responses

The $\Delta F/F$ traces shown throughout the manuscript (e.g., Figures 1G, 2C, 5B, 6B and 7B) exhibit several features that raise questions about signal processing and the suitability of calcium imaging for certain timing-sensitive claims:

Post-stimulus suppression: A consistent dip in $\Delta F/F$ below baseline following evoked peaks appears in both widefield and two-photon recordings. The suppression size is remarkably larger than the evoked activity. This pattern is uncommon, even with faster indicators like jGCaMP8m, and could result from baseline inflation or decay overlap across high-frequency stimuli (e.g., 5 Hz).

Pre-stimulus rise: The sound-evoked activity ($\Delta F/F$) begins to increase prior to stimulus onset in several examples. This observation requires further clarification and investigation. The manuscript should address: 1. Is there a potential timing misalignment between stimulus delivery and the recording system that could account for this apparent pre-stimulus activity? 2. Have the authors ruled out technical factors such as hardware synchronization issues or processing delays in data acquisition? 3. Could this observation reflect a physiological anticipatory response related to the experimental design, such as stimulus entrainment or expectation effects? 4. What steps were taken to ensure proper baseline normalization, particularly with respect to potential contamination from preceding stimuli?

Baseline strategy: $\Delta F/F$ was calculated using the median fluorescence over an 8-second window. While this approach reduces trial-by-trial noise and global drift, it also incorporates activity from multiple preceding stimuli, especially at high presentation rates. This could inflate baseline values and potentially produce apparent "responses" during omissions that reflect returns to true baseline rather than stimulus expectation violations.

These signal dynamics directly affect the interpretation of the timing, magnitude, and spatial localization of omission responses—the central focus of the manuscript. From a conceptual standpoint, the interpretation of these responses as "prediction errors" assumes that the neural signals are temporally precise and reflect a violation of an internal model. However, without behaviorally relevant prediction, and in the absence of a task or outcome, it is difficult to rule out alternative mechanisms such as entrainment-related rebound or passive stimulus filtering, which could produce similar dynamics in the absence of cognitive prediction.

In this context, I also wonder whether the choice of imaging methods over electrophysiology introduced tradeoffs that affect some of the conclusions. The widefield and two-photon approaches clearly offer valuable spatial insights, particularly in identifying omission response localization in ORR (likely TeA). However, for capturing precise, rapid dynamics such as SOA dependence, ramping, and double-omission responses at 200 ms intervals, the temporal limitations of calcium imaging may pose a challenge. Even with GCaMP8m, the intrinsic integration and decay properties limit the ability to resolve sub-200 ms timing features, especially without deconvolution.

I raise this not to question the value of imaging, but to ask whether the authors might consider:

Showing representative raw fluorescence traces and/or overlaying $\Delta F/F$ with the baseline F_0 to clarify the impact of preprocessing,

Exploring $\Delta F/F$ computed on a trial-by-trial basis, or including deconvolved activity estimates, to better match the temporal structure of the experimental design, and possibly discussing (or incorporating) complementary electrophysiological recordings, even in a limited fashion, to validate key timing-based inferences (e.g., omission latency, SOA scaling, ramping structure).

Omission vs. Offset Controls

In Figure 5, the authors attempt to distinguish omission responses from offset responses by comparing activity in a 200 ms SOA sequence (with omissions) to that following sequence termination in a 400 ms SOA stream. However, these conditions differ not only in the presence or absence of a final tone, but also in their temporal context—specifically, the rate of preceding stimulation, the level of neural adaptation, and the strength of temporal entrainment. These differences could confound any conclusions drawn from the comparison. For instance, fast rhythmic input (as in the 200 ms condition) can induce phase-locked neural activity that gives rise to so-called “echo responses” (<https://doi.org/10.1038/s41598-017-18465-w> ;) — rebound-like transients that occur at expected times even after stimulus cessation, potentially reflecting rhythmic resonance rather than prediction error (<https://doi.org/10.1523/JNEUROSCI.1642-19.2019>).

Additionally, the interstimulus response shown in the 400 ms SOA condition (Fig. 5B), which the authors use as an offset control, appears to persist well into the time window where an omission response would occur in the 200 ms condition. This temporal overlap suggests that what is interpreted as an omission response may in fact ride on ongoing offset-related activity, particularly since the omission window occurs just 200 ms after the prior stimulus. This possibility deserves further attention, as it directly bears on the specificity of the omission response as a prediction error signal.

More broadly, this raises a conceptual concern: how do we differentiate between a predictive signal that reflects expectation violation, versus a biological rhythm echo or offset-like signal that appears due to entrainment or rebound following regular input?

The double-omission control helps to address some of these concerns, but its interpretation may be complicated by the same $\Delta F/F$ signal dynamics noted earlier — especially the strong post-stimulus suppression and potentially inflated baselines that arise in high-frequency stimulus contexts. These dynamics may produce rebound-like features that resemble offset or echo responses, further challenging the distinction between prediction-driven and passive neural activity. Clarifying how baseline and decay dynamics were handled in these controls, and providing raw or minimally processed traces where feasible, would greatly aid in interpreting the origins of these signals.

Jittered SOA Condition and Prediction

The jitter control in Figure 6 is conceptually valuable and well-designed in intent, but I found it difficult to interpret the presence of any omission response under this condition.

The jittered sequence uses a pseudorandom set of SOAs (100–300 ms), eliminating consistent temporal regularity. From a predictive coding perspective, one might not expect strong omission responses in the absence of a stable rhythm to generate precise temporal predictions.

The authors do report reduced ORs under jitter (Fig. 6B), but not abolished. The presence of even reduced omission responses in this context raises a question: are these signals truly reflecting a top-down temporal prediction error, or do they reflect bottom-up adaptation or rebound dynamics that are less sensitive to fine-grained regularity? Without a behavioral marker of prediction or evidence of hierarchical feedback, it is difficult to conclude that these signals represent cognitive prediction violations rather than statistical learning or passive entrainment. Additionally, the same $\Delta F/F$ signal dynamics described above (e.g., suppression, baselining issues) may behave differently under regular vs. jittered timing, potentially exaggerating this apparent drop.

If the omission response reflects a rebound or offset-like phenomenon, this could confound the intended interpretation of the jitter control as evidence for prediction-specific processing.

Statistical Framework

A few statistical issues are worth revisiting:

MANOVA (Fig. 2H) assumes multivariate normality and independence, which is difficult to justify in calcium imaging. Non-parametric alternatives may be more robust here.

The data are hierarchically structured (neurons within animals), yet most tests treat each neuron as independent. Animal-level averaging or mixed-effects models would more accurately reflect the design and reduce pseudo-replication.

Behavioral Measures (Figure 4)

The inclusion of facial motion and pupil responses is an important strength, but I would encourage a more cautious interpretation:

Facial motion increases after omissions but is time-locked to the next sound, not the omission itself. This makes it difficult to argue that the movement reflects prediction error rather than rebound excitation or adaptation. There is also no reported correlation between motion energy and neural responses.

Pupil dilation peaks ~500 ms after the omitted stimulus—well after the neural omission response. This is compatible with a general arousal response, possibly LC-NE-mediated, but not direct evidence for predictive monitoring. Moreover, the dilation overlaps with responses to subsequent stimuli, further complicating interpretation. These signals may be better framed as nonspecific arousal correlates rather than behavioral readouts of predictive timing, particularly given the lack of task engagement.

(Remarks on code availability)

(Remarks to the Author)

This manuscript by Peters, Cai et al. describes results of experiments designed to investigate the neural representation of sound omissions (unexpected absences from an otherwise statistically predictable sound sequence) in the auditory cortex of awake mice. Neural responses to unexpected omissions are of great interest to sensory neuroscientists because they arguably represent the purest form of evidence for statistical learning and/or sensory prediction; there is no stimulus present during an unexpected omission and yet the omission evokes neural activity. Omission responses have been widely demonstrated in humans using EEG and also, in a few studies, in animals including rats and mice. However, the neural mechanisms underlying omission responses and their localisation within the brain have remained largely open questions. The answers to these open questions have profound implications for prominent theories about "predictive coding" in the brain, with potential to help resolve debates about the extent to which neural responses in different cortical and subcortical regions represent stimulus features, sensory predictions, prediction errors, or something else.

Here, experiments were conducted in awake passively listening mice, while neural activity across primary and secondary fields of auditory cortex was monitored using widefield imaging and, at the single-neuron level in more targeted fields of view, using two-photon imaging with a fast calcium indicator, jRCaMP1e. Facial movements were measured during the experiments, providing a spontaneous behavioural measure of arousal. Multiple stimulus paradigms were used to rule out alternative explanations for the results and to test predictions of prominent theories about neural representation, such as that omission responses signal sensory predictions or prediction errors.

The manuscript makes at least three major contributions to systems neuroscience:

(A) Results clearly demonstrate the existence of a secondary region of the auditory cortex in mice (pTeA) in which neurons with strong omission responses are clustered. There are multiple well-designed controls that effectively rule out alternative explanations, and some of these controls are quite novel and innovative in their own right (e.g. the second-omission control, which could also be useful for investigating omission responses in human EEG studies).

(B) Intriguingly, the temporal characteristics of the omission responses observed in the mouse brain are inconsistent with some theories of predictive coding. In particular, the observed responses ramp up monotonically following the unexpected omission rather than following the expected timecourse of the omitted stimulus, implying that the responses represent something more like an "integrated" prediction or prediction error signal. This result provides an important new constraint for theories of sensory representation in the auditory cortex, and for analysis of "mismatch responses" and other deviant responses in all sensory modalities.

(C) Finally, the manuscript serves as an excellent early illustration of the power of fast two-photon imaging techniques to address questions about neural representation on ~10ms timescales. The authors used a relatively new and very fast calcium indicator (jRCaMP1e) and rigorously minimised the use of deconvolution and filtering, and thereby provide a very robust demonstration that calcium imaging can be used effectively to explore questions about temporal representation of sound signals that would previously have been robustly accessible only using electrophysiological techniques.

The manuscript is also very well-written. The authors should be particularly commended on the thoughtfulness of their Discussion section, which carefully lays out the evidence against possible alternative explanations for the key observations, and explores a wide range of implications of the results for systems, cognitive and computational neuroscience.

MAJOR COMMENTS:

1. Data on the laminar distribution of omission responses vs standard responses is less convincingly presented than the other results. In particular, Figure 3 is not fully explained. What do the coloured vertical strips in the top lineplots of Figure 3A represent? What are the axes of the matrix plots in Figure 3A? More critically, why is the "omission response" line in the Figure 3C not normalised to 100% at 100um, given that the x-axis label says "Response rel. OR at 100um"? While the other figures in the manuscript were generally very convincing and largely self-explanatory, Figure 3 is not convincing in its present form.

2. Several controls provide compelling evidence demonstrating that activity in the ORR is related to omissions rather than offset responses (e.g., comparison between omissions at 200ms SOA and expected sounds at 400ms SOA, and observations regarding the double-omission response). However, the claim that the ORR can be clearly distinguished topographically from regions with offset responses could be better supported. Please provide more information about exactly how offset response magnitude in Supplementary Figure 3 is defined. Peak $\Delta F/F$ over what time interval after offset? Robust identification of offset responses is not as easy as for onset responses because an excitatory offset burst can be difficult to distinguish from a slow decay in sustained activity. There's a lot of overlap in the onset and offset response maps in Supplementary Figure 3; this is potentially unsurprising as a lot of auditory cortex neurons have responses to both transients. However, another possibility is that the offset response measure is slightly confounded here with the onset response measure, because a cell with high sustained response that decays slowly following sound offset might show up as having both a large onset response AND a large offset response, even in the absence of a "true" offset response burst. What would maps of the difference in $\Delta F/F$ between the offset period and onset (or onset+sustained) period look like? More generally, how do the authors know that their offset response measure is capturing "true" offset responses rather than slow decays of sustained activity?

3. Also regarding maps in Supplementary Figure 3 (and elsewhere): There seems to be a very prominent outlining of blood

vessels in the response latency, response modulation, onset response, offset response and omission response heat maps. Why might this be?

4. The dependence of omission responses on SOA was non-monotonic with SOA (Figure 5O). Also, the dependence of omission responses on the number of previous presentations of regular stimuli was slightly non-monotonic (increasing to 40-70 trials then slightly decreasing). Both of these observations would seem to suggest that entrainment/oscillation phenomena might be a more accurate and parsimonious explanation for the observed omission responses than integrated prediction or prediction error. Granted, these explanations are not possible to disentangle cleanly, but the non-monotonicities provide an intriguing avenue for further investigation. Do measures of entrainment in the responses to non-omitted stimuli show the same profile as a function of SOA or number of previous stimuli as the omission responses do? If so, this might be good evidence for the entrainment hypothesis --- or at least, good evidence for circuit-level constraints on the more cognitive predictive-coding interpretations, which would not predict non-monotonicity with SOA or number of previous stimuli.

MINOR COMMENTS:

Abstract: "Animals were aware of omissions..." Aware is something of a loaded word and should be avoided. It is possible that the arousal responses measured in spontaneous behaviour are entirely subcortical and only minimally relevant to "awareness".

Abstract: I think the abstract could be slightly improved to highlight the significance of this paper more clearly. In particular, one of the greatest strengths of this paper is the careful comparison of different stimulus paradigms (varying SOAs, jitter, number of omissions etc) to rule out offset responses or other possible explanations for the observed omission responses. This is summarised mainly in the sentence about omission responses being "temporally distinct" from offset responses, which doesn't really do the authors' efforts here full justice.

Figure 4, panel O: This plot is described in the legend as showing a "low-pass dependence" of OR on SOA, but in fact it is bandpass with peak at 200ms. Why might this be? More explanation or at least speculation here would be helpful. The 200ms SOA almost seems a bit of a "sweet spot". In fact for SOAs above 300ms it seems that the omission response is plateauing. Could this reflect an interaction with brain oscillations, or (perhaps equivalently) with short-term synaptic plasticity timescales?

Figure 6: It's very intriguing that the effect of sequence jitter on the omission responses was so minimal once the impact of sequence jitter on baseline activity was taken into account. This observation seems itself worth a bit more attention in the Discussion.

Line 57: "tone height" -> tone frequency

Line 59: "predictive processings" -> predictive processing

Line 74: "Calcium" should be lowercase

Discussion: Generally the discussion is excellent (especially the subsections on alternative explanations for omission responses and implications for predictive coding). However, the first subsection on relation to previous studies could be improved; it doesn't flow very well across some seemingly disjointed paragraphs. For example the paragraph about whether omissions are regularly or randomly selected (lines 402-405) feels a bit disconnected from everything else, and it's not really made clear what random or regular selection of omissions means.

The Discussion would also benefit from a bit more information about how the areas known to connect to ORR might contribute to generation of the monotonically ramping omission responses (and/or changes in baseline activity with jittering of regularity, which was also a very intriguing observation in itself).

(Remarks on code availability)

Version 1:

Reviewer comments:

Reviewer #1

(Remarks to the Author)

The authors have fully addressed all my comments.

(Remarks on code availability)

Reviewer #2

(Remarks to the Author)

The authors have made substantial efforts addressing my concerns by providing more analyses and clarifications. The spatial findings—TeA localization, layer-specific distribution, and hierarchical organization—are robust and represent valuable contributions to understanding predictive processing. However, claims about temporal precision throughout the manuscript exceed what the methods can establish. The authors themselves acknowledge: "While it cannot be concluded with certainty..." (response line 190-191) and note "baseline level is hard to determine during continuous stimulation" (response line 216). Given 2-photon imaging at 30 Hz (33ms resolution) and jGCaMP8m decay of ~40ms, they observe temporal structure at this scale but cannot prove neural computation precision versus measurement ceiling.

I would suggest the following text changes:

Line 15-16 (Abstract): Change "Neuronal responses to the omission of expected sounds were precisely timed" to "Neuronal responses to the omission of expected sounds were time-locked to expected stimulus onset"

Lines 82-83, 130, 252 (Results): Replace all instances of "precisely timed" with "time-locked to expected onset"

Lines 408-414 (Discussion): After "temporally precise (10s of ms) mechanism," add: "While jGCaMP8m provides improved temporal resolution, the precise timing of predictive computations at the millisecond scale remains to be confirmed with electrophysiology"

Line 475 (Discussion on offset responses): Change "effectively ruled out" to "substantially reduced the likelihood" (authors acknowledge: "While not impossible, we consider this rather implausible" - response line 484)

Lines 127-128, 157-158 (OR magnitude): Add caveat that magnitude estimates depend on baseline choice during continuous stimulation

Lines 89-92 (Abstract, behavioral): Change "reflected in behavioral responses" to "correlated with behavioral changes"

Lines 583-587 (Discussion, limitations): Add that active paradigms are needed to distinguish prediction monitoring from arousal responses (authors state: "only an experiment involving active behavior can provide more clarity" - response line 846)

With these revisions, I recommend publication.

(Remarks on code availability)

Reviewer #3

(Remarks to the Author)

The authors have done a very thorough job of addressing my comments and I have no further suggestions for revision. Thank you for the detailed explanations and additional analysis.

Regarding the authors' query about whether their new Figure S9 addressing questions about entrainment should be added as a main figure --- I think it's probably best as a supplementary figure given the level of detail, but it is a very nice addition nonetheless.

(Remarks on code availability)

**Response to the reviewers' comments for the manuscript NCOMMS-25-15638-T**

"The representation of omitted sounds in the mouse auditory cortex"

We would like to thank the reviewers for their thoughtful and constructive comments, which we
address in detail below, including several new figures. Changes to the manuscript are highlighted
in blue font.

**Reviewer #1 (Remarks to the Author):**

**This is a very interesting manuscript combining widefield and 2-photon imaging calcium**
**imaging as well as pupillometry and facial movement analysis to provide novel insights**
**into the neural representation of omitted sounds. The most significant aspect of the work**
**is the demonstration of the localisation of omission responses to a particular region of the**
**temporal cortex, which will be extremely valuable to those intending to dissect the circuitry**
**underlying these omission responses.**

**R:** We would like to thank the reviewer for this highly encouraging assessment of our work. We
fully agree with the implications this holds for the future analysis of the circuitry, which we are
planning to tackle in the near future.

**The authors produced a solid piece of work which will certainly have an impact beyond**
**the field auditory neuroscience. The methodology is sound and the conclusion are well**
**supported by the evidence.**

**I only have some (very) minor comments.**

**R:** Thanks for providing such detailed comments, which helped us to iron out these remaining
typos/errors!

**Figure S4: put orientation bars on photos**

**R:** Thanks, added and figure updated.

**L.59: processings > processing**

**R:** Thanks, changed!

**Figure 1H: panel label missing**

**Figure 1I: incorrectly labelled as 1H**

**R:** Thanks for mentioning this potential confusion. We in fact had intended it to be layouted like
this, i.e. with the two panels receiving a single letter, as the description of the area and the time-
course are dependent on each other, and neither sequence with separate letters could be
unambiguous (i.e. in one first shows the time traces, the areas are not yet defined, and if one first
shows the areas, then it is not clear at which time the activity is shown). But we noticed that this
was incorrect in the results text, which we have now corrected to match the figure and its caption.

**Figure S2E: I don't think I understand panel S2E and I'm wondering whether it adds**
**much/anything to what's already shown in S2D.**

**R:** Yes, we agree, this was not sufficiently clear yet. The data in S2E is an example from a single-
animal, while S2D shows the shift of the PureTone ORR center, with respect to the NB ORR
center, across animals. We have added more information to the caption and changed the Panel
title of S2D to clarify this.

**Figure 2B: The ORR demarcated in 2B looks much larger than what was identified as ORR**
**in S2D and would seem to encompass parts of A1. Or am I misinterpreting something here?**
**I guess, to some extent, this may be explainable by differences in scale perhaps? Are all**
**data in Figure 2 from a single animal?**

**R:** Thanks for noticing this! We have gone through the code and noticed that the threshold for
demarcating the Omission response area was accidentally set to a value that was lower than in
all other plots and given in the methods. We have corrected this mistake and updated the figure,
which now shows that the 2p ROI is encompassing a substantial part of the ORR.
Panels F, G, H, and I in Figure 2 present data pooled from individual cells across multiple (5)
animals. While this was mentioned in the figure caption, we have further clarified it there, and also
clarified it in the associated Results sections.

**L.155: more steep > steeper**

**R:** Thanks, changed!

**L.1008: delete 'across'**

**R:** Thanks, done!

**L.372: an > a**

**R:** Thanks, done!

**L.393: times > timed**

**R:** Thanks, changed!

**L.402: typically > typical**

**R:** Thanks, changed!

**L.501: receptive properties > receptive field properties**

**R:** Thanks, added!

**L.524: suggest > suggests**

**R:** Thanks, changed!

**L.669: used image > used to image**

**R:** Thanks, changed!

**L.778: delete 'but'**
R: Thanks, removed!
**L.785: It is somewhat unclear how the authors dealt with the neuropil contamination. Was the neuropil signal subtracted in full (F-Fneu) or was the neuropil signal scaled before subtraction (i.e. F-0.7*Fneu) as suggested by Chen et al., 2013 (GCaMP6) and Zhang et al., 2023 (GCaMP8). The sentence starting with "was empirically selected..." is missing at least one word at its beginning and it is not clear at all what the authors are trying to say here.**
R: Thanks for pointing this out. We indeed followed the typical approach in the community to subtract a scaled version of the neuropil signal. However, we found that the correlation between the neuropil signal and the cells was typically rather small (~0.2), and we set the scaling factor to a value <0.7, namely in the range from 0.3-0.5. This choice has been clarified in the methods section and the formula has been more explicitly described. Using a value of 0.7 led to qualitatively similar results.
**L.788: "then for further": a word seems to be missing here.**
R: Thanks, 'collected' added!
**L.893: 'region' missing after 'responsive'**
R: Thanks, added!
**L.974: centrum > centre?**
R: Thanks, corrected!
**L.1020: I think I worked out what is displayed here (Fig 3A) but I'm not absolutely certain. This panel legend, like many others, provides the authors' interpretation of the displayed data rather than a detailed and neutral description that would allow the reader to arrive at their own interpretation. What are the axes in the top panels (there are no scale bars)? What is the green vertical line?**
R: Thanks for pointing this out, there were indeed differences in the visual style between this figure and the others, which we have now resolved. In particular we have added scale bars for time and dF/F, use the same style for the omission and added a label for the stimuli/omission (vertical bars). Also, the green bars, which are time-indicators for stimulus or omission, have now been changed to match the color scheme for those two cases, and visually connected to the panels that are displayed below. These changes have been clarified in the text and in the figure caption. Regarding the level of interpretation in the captions, we have carefully revisited all figure captions, removing formulations that were suggestive of an interpretation, rather than a description. However, for clarity of presentation we still refer to the elevated activity in the omission period as Omission responses.
**Figure 5J: I don't understand what is displayed here.**
R: We agree that this analysis was not sufficiently explained in the manuscript. We have now adapted the corresponding sections in the Methods ("Spatial clustering of neural responses to double omission") and the Results to clarify it. We would also like to take the opportunity to clarify

it here directly for the reviewers: The motivation to perform the cluster analysis was to highlight
regions which show a heightened response to the second omission in sequence. Since the double
omissions could only be integrated less frequently than the single omissions (as otherwise, the
animal could have expected them), the corresponding regions were less robustly defined than the
omission region. The clustering was performed on one-dimensional response trace profiles from
double omission, and the resulting cluster labels were projected back onto the craniotomy for
spatial visualization.

**Reviewer #2 (Remarks to the Author):**

**This manuscript investigates neural responses to omitted auditory stimuli using widefield**
**and two-photon calcium imaging in mice, identifying spatially localized omission**
**responses and proposing their relevance to predictive processing. The data are extensive**
**and the approach is ambitious, combining neural recordings with behavioral measures**
**(motion energy and pupil size) and a series of control manipulations. However, several**
**technical and interpretive concerns—particularly surrounding the $\Delta F/F$ signal**
**processing—raise questions about the robustness of some core findings. The intention**
**here is to highlight areas where clarification or re-analysis could meaningfully strengthen**
**the conclusions.**

**R:** We thank the reviewer for taking a constructive approach to the review process, in particular
in the context of recent methods, which bring new possibilities (much improved SNR and temporal
resolution), but we also agree that their introduction to a field requires particular caution, which
we provide below in the form of additional analyses to address the reviewers' specific concerns.
As there is partial overlap with questions raised by R3 (on the relation to entrainment) and within
the suggestions of this reviewer, we partially point to other sections of this letter. Generally, we
would like to emphasize here that we mainly provide data that can critically inform candidate
theories of predictive processing, and carefully discuss in how far they are compatible with the
different frameworks, for example 'Predictive Coding'. Instances where descriptions could have
been suggestive, we have carefully adapted to be more neutral, while maintaining an accurate
description of our findings.

**Calcium Signal Processing and Temporal Resolution in the Interpretation of Omission**
**Responses**

**The $\Delta F/F$ traces shown throughout the manuscript (e.g., Figures 1G, 2C,5B,6B and 7B)**
**exhibit several features that raise questions about signal processing and the suitability of**
**calcium imaging for certain timing-sensitive claims:**

**Post-stimulus suppression: A consistent dip in $\Delta F/F$ below baseline following evoked**
**peaks appears in both widefield and two-photon recordings. The suppression size is**
**remarkably larger than the evoked activity. This pattern is uncommon, even with faster**
**indicators like jGCaMP8m, and could result from baseline inflation or decay overlap across**
**high-frequency stimuli (e.g., 5 Hz).**

**R:** We noticed this pattern as well and have spent considerable time investigating it as to our
knowledge there is no published data for our technique (GCaMP8m in ACX), the results of which
we will gladly share with the reviewer. Our first question was, whether the negative slope after the
stimulus-related activity peak is only an artifact of the indicator. At the beginning of the trial (see
figure R1 below, grand average across all animals), the strong negative downslope appears to
only develop over multiple presentations of the same stimulus, associated with a speedup of the
excitatory peak as well. This observation, in combination with the indicator's dynamics (~2.5 ms
rise time and 50% decay time ~40 ms see Extended Data Table 6, (Zhang et al. 2023)) causes
181 us to conclude that indicator dynamics do not explain the observed dip. This leaves physiological
explanations.

The first possibility would be forward suppression, i.e. short-term synaptic depression at
thalamocortical and intracortical synapses produces suppressed firing that last on the order of
tens to hundreds of milliseconds; when stimuli repeat at ~5 Hz, recovery is incomplete and
suppression accumulates across the train, producing a below-baseline period after offsets (Wehr
& Zador 2005; Chung et al. 2002; Ulanovsky et al. 2004). However, in our data we see that the
post-stimulus dip is very comparable between the 200SOA and 400SOA conditions, while the OR
is nearly extinguished for 400SOA (Fig. 5B & O, Fig. S10).

While it cannot be concluded with certainty, to us this suggests a contribution of rapid
inhibition, which builds up with the sequence, effectively leading to stimulus entrainment (see R3,
Main Point 4 for a detailed discussion of entrainment). We have discussed these response shapes
on the Slack channel with multiple researchers, among them Loren Looger (Janelia), whose group
developed jGCaMP8m. The conclusion from this conversation was that the observed time-scales
are consistent with neural activity, see Zhang et al. 2023, Fig. 4d (excitatory neurons in VI) with a
50% decay time of ~70ms, and members of this forum agreed that inhibitory inputs could further
shorten the observed dynamics. The hypothesis of inhibitory input would also be consistent with
the relative activation times of excitatory and inhibitory cells in the auditory cortex, with excitation
preceding inhibition (see Fig. 3F from Feigin et al. 2021, reproduced below in Fig. R2 for the next
point).

Indeed, multiple single-stimulus electrophysiology studies show that an isolated tone or
brief sound can produce a transient suppression of spontaneous firing below baseline after the
initial evoked response (Willmore & King, 2022) — durations typically range from tens to a few
hundred milliseconds depending on brain level (subcortical/cortical) and cell type (Voytenko &
Galazyuk, 2010; O'Connell et al., 2011; Sadagopan et al., 2023). Therefore, we conclude that the
indicators' fast dynamics resolve a real physiological phenomenon, most likely experience-
dependent inhibitory sharpening.

Regarding the suggestion of baseline inflation: We have investigated this and there
appears to be an adaptation of the overall brightness over time, which starts already from the
beginning of the trial (see Figure R1), even before acoustic stimulation begins (for more details
on this see the Figure R3 in this letter, which details baseline adaptation in silent and stimulus
driven trials). We think that this baseline reduction is in fact likely indicator-related, and mainly
does not originate from an adaptation in neural activity, as it also occurs in silence and persists
after the termination of the stimulus (although it appeared to recover between recording sessions),
and has been observed before (see Fig. R1b below). This motivated our choice of using a
dynamic baseline. However, we agree that the exact level of this baseline is hard to determine
during continuous stimulation as is the case in an oddball paradigm (see Figure R3 for more
details on baseline choices below). Therefore the absolute size of the negative part of the
response after the stimulus might indeed be increased by the choice of the dynamic baseline,
however, it would be important to note that the polarity can still likely be correct (see Fig. R1c
below, which shows neuronal response profiles from the mouse auditory cortex recorded using
electrodes, adapted from (Jamali et al., 2024)).

Regarding the reviewer's suggestion of decay overlap: We are confident that this
possibility can be ruled out, since an overlap would have to extend over periods exceeding a
single SOA, i.e. >200ms. On the one hand, the jGCaMP8m response kernel is already quite
decayed at this point (see Zhang et al. 2023, Fig. 4d), and the response components that we

observe in between the last response and the next negative dip show faster dynamics, which
would raise the question why such a potential residual (negative) decay was not visible in between
the stimuli, and would then materialize after the following response.

In summary, we think the most likely explanation of the negative dips is a combination of
a limited overestimation of the baseline in combination with an actual buildup of inhibitory input
and would argue that the use of previous calcium indicators (e.g. GCaMP6s/f) would have
obscured such dynamics. We would like to emphasize that this does not affect the differences we
observe between ORR and PR for omissions, which is the main result of the present manuscript.
We have added the figure below to the manuscript (Supplementary Fig. 8) and detail these
aspects now in the Methods.

**Figure R1: Stimulus specific adaptation and gradual response shape morphing occurs at the**
**beginning of the sequence.** Data in this plot is derived from the median over both trials of all 10 mice for
the 32kHz Omission paradigms, for ORR (red) and PR (blue), aligned to recording start. A large and
relatively undefined response occurs for the first stimulus in the PR, accompanied by a slower excitatory
modulation (~0.5s half-decay time) across both areas. With an increasing number of stimuli, a fast baseline
undershoot develops after the stimulus onset response (see Fig. 1G for mean response shape after
sequence habituation has occurred).

[FIGURE REDACTED]

**Figure R1b: Adaptation to a stimulus sequence adapted from Li et al., 2017**
**(doi.org/10.1093/cercor/bhx079).** These recording are also from the auditory cortex of mice, using a
different, slower indicator and a slower stimulation rate (0.5 Hz), however, also here a reduction in the
baseline is observed.

[FIGURE REDACTED]

**Figure R1c: Adapted from Figure 3 (Figure supplement 3, Panel D),**
**(<https://doi.org/10.7554/eLife.102702.2>).** This paper provides electrophysiological recordings from
populations of neurons in the mouse auditory cortex, in response to 5 sounds with similar SOA (250 ms).
Neurons were here clustered into groups by different response types. The first cluster (left) shows strong
reduction of activity in response to the stimulus, while the second one shows below baseline reduction after
the 4th and 5th stimulus, and the third one (right) shows adaptation. In our widefield imaging these clusters
would be combined and might give rise to the observed composite waveform.

**Pre-stimulus rise: The sound-evoked activity ($\Delta F/F$) begins to increase prior to stimulus**
**onset in several examples. This observation requires further clarification and**
**investigation.**

**R:** We have carefully checked all figures (specifically Fig. 1C/G, 5B/H, 6B/H, 7B) but could not
find cases in which the actual response occurs before the stimulus. We think there are two
reasons for why responses may sometimes appear to occur early, which we would like to
exemplify with Fig. 2C (2p imaging).

Firstly, the first sample after stimulus onset shows the peak response, which may appear
as too early. Given the response delay in the mouse auditory cortex is ~ 5 -10ms (see e.g. Feigin
et al. JNeurosci. 2021. Fig. 3F, reproduced below as Fig. R2) and the ~ 2.5 ms onset time (50%
activation) of the jGCaMP8m indicator (Zhang et al. Nat. Methods, 2023, Fig. 1D, and Extended
Data Table 6), it is in fact to be expected that a substantial part of the onset response of a neuron
falls into the first 2p imaging bin which is 33 ms long. Hence, while at first glance seemingly too
fast, the fast dynamics of AC responses and the indicator response are in line with our
observations.

Secondly, sometimes a rise is observed already before the next stimulus. This is correct,
and has also been puzzling to us at first. In addition to the technical checks we performed (see
below), we have compared extensively across cells and paradigms, and have reached the
conclusion that these rises are due to either recovery or anticipatory dynamics following the
preceding stimuli. We base this conclusion on the fact that we find consistent timing at stimulus
onset for many different paradigms, and also the dynamics that we observe during an omission,
as well as the positive inflection in the response size that happens in the first bin after the next
stimulus has been presented (see in particular the single cells shown in Fig. 2C bottom, light
blue).

Thirdly, we used to center the dots in the 2p results at the center of the time window. While
this would usually be considered correct in principle, it contributes to the observation of the
reviewer, as information from the future (up to $33 \text{ ms}/2 = 16.5 \text{ ms}$) can add to the data shown at
each point (depending on where a neuron lies in the field of view, and the subsample correction
we performed, see Methods). To avoid this confusion we have now plotted the data always such
that a measurement point represents the data that has been collected in the dT before it, i.e. on
average, this shifts the plot to the right by 16.5 ms. We would like to ask the reviewer to revisit
Fig. 2C and check whether this has resolved the concerns.

We have made additions to the Methods description and updated the 2p figures
accordingly, in regards to the points mentioned above.

[FIGURE REDACTED]

**Figure R2:** Figure 3F (from Feigin et al. 2021), which shows the rapid onset latencies. Note also
that inhibitory interneurons follow a bit later (right plot) and peak between 50-100ms.

**The manuscript should address:**
**1. Is there a potential timing misalignment between stimulus delivery and the recording**
**system that could account for this apparent pre-stimulus activity?**
**2. Have the authors ruled out technical factors such as hardware synchronization issues**
**or processing delays in data acquisition?**

**R:** The two points above are naturally very important in any experiment and related to each other,
and we have very carefully designed and validated the stimulus generation and alignment to the
recorded data to make sure that there are no temporal misalignments. Below we provide a list of
the checks that we have performed to ensure this (already before submission of the original MS):

- ● For the widefield data, we originally started with a rolling-shutter sCMOS camera, which
led to different timing per line in the image (data not included in this manuscript). To ensure
identical timing for all pixels in an image, we transitioned to the global shutter camera that
delivered the data for the present study. For the 2p data, due to the nature of the laser-
scanning process, this issue is unavoidable. To correct for it, we time-shift pixels on a
given line of the image by the expected delay, i.e. for 30 Hz this corresponds to $33\text{ms}/512$
$= 0.064\text{ms}$ (differences between dots on a line are negligible).
- ● Both widefield and 2p systems are hardware synchronized to the stimulus, i.e. specifically:

- ○ The widefield camera receives frame acquisition triggers which are generated and
rerecorded by the NI DAQ device, which functions as the main clock in the
experiment. The camera also sends acquisition triggers back, which are recorded
on an analog channel by the same DAQ device. The same device also generates
the sound starting at a digital trigger that is also recorded again on an analog
channel. In this way, all these signals are precisely synchronized via the analog
channels recorded in parallel (at a resolution of 0.1 ms).
- ○ The 2p scope runs on an intrinsic clock but sends frame acquisition triggers, which
are recorded on an analog channel as well, and are then post-hoc aligned to the
start trigger for the sound output (same as above in widefield).
- ● In widefield imaging, frame-losses can occur occasionally due to the high frame rates. We
correct for these posthoc, by comparing the frame triggers, frame acquisition triggers and
recorded frame times. Missing frames (which are ~0.01% of all frames) are replaced by
the previous frame, thus avoiding any shifting forward in time, as feared by the reviewer.
In 2p imaging, we have not encountered any frame losses.
- ● The spectral correction of our speaker introduces a delay of a few milliseconds. This delay
is constant and is already corrected for during stimulus production, by forward-shifting the
stimulus by this delay, such that the actual onset of the sound happens at the intended
time (i.e. due to the convolution with our inverse filter kernel and the filtering of the
amplifier/speaker.
- ● We have validated the entire synchronization of the recording system by recording the
sound and filming the membrane of the speaker, which showed precise temporal
alignment.

In summary, we are certain that there are no unaccounted delays in the system between
stimulation, video and neural recording in both widefield and 2p imaging. We have added a
shortened version of these checks to the Methods.

**3. Could this observation reflect a physiological anticipatory response related to the** 348 **experimental design, such as stimulus entrainment or expectation effects?**

**R:** While - as we argue above - this effect is likely smaller than initially perceived by the reviewer
(we would also like to ask the reviewer to reinspect the updated plots in this respect), we
nonetheless agree that the inter-stimulus dynamics of the neural response might represent other
processes than just residual neural/indicator activity from the previous stimulus. One of R3's major
points (#4) was also about entrainment, we would like to point R2 to the detailed response there.
In brief, our entrainment analysis indicates that some entrainment to the stimulus occurs indeed,
with the bipolar response size to the stimulus growing at the beginning of the sequence, after a
short phase of adaptation of the positive response. However, we find no evidence that the
omission response's shape entrains to a similar shape as the one of the stimulus response.
Instead, we think that the rise towards the next stimulus is already the beginning of the omission
response, which - consistently with this interpretation - is also larger in the ORR compared with
the PR. In that sense we think it is most consistent with an expectation effect. We now mention
this interpretation in the discussion, and have substantially expanded our analysis of entrainment
including a new figure (Fig. S9).

4. What steps were taken to ensure proper baseline normalization, particularly with respect to potential contamination from preceding stimuli?

R: As discussed above, the dynamic baseline might be overall sitting a bit higher than the actual baseline, which would thus have shifted the response kernel to somewhat more negative values. Given the length of our dynamic baseline (8s) in relation to the SOA (0.2s), there should be no other contamination aside from this shift as the average value is formed over the span of 40 stimuli. The 8s window was chosen conservatively from the outset (i.e. only coarse drift is removed and not all local baseline variability is eliminated, see Fig. R1), in order to err on the side of caution with respect to the response shapes. We provide additional analyses and details regarding this question together with the following point, which concerns a related question.

Baseline strategy: $\Delta F/F$ was calculated using the median fluorescence over an 8-second window. While this approach reduces trial-by-trial noise and global drift, it also incorporates activity from multiple preceding stimuli, especially at high presentation rates. This could inflate baseline values and potentially produce apparent “responses” during omissions that reflect returns to true baseline rather than stimulus expectation violations.

R: Based on the arguments above as well as the additional analyses (see following Figure), we are certain that the omission responses do not reflect a return towards a more positive baseline. As we show below, the baseline is actually sitting lower - rather well approximated by our moving baseline - and the omission response first increases relative to this baseline before returning to it (see Figure R3H below). In fact, this was one of our main concerns initially as well, however, the additional analyses clearly indicate that the omission response has a specific temporal onset, but extends in time longer than predicted by typical predictive coding theories. We will go through the rationale for our baseline choice detail below. Importantly, we would like to emphasize that *no baselining was performed in the 2p recordings and the shape of the omission response was quite comparable* to the one in the widefield case in the omission responsive region.

We chose a dynamic baseline to counteract long-term dynamics in the recorded signal that are rather unrelated to the short-term neural response, relating mostly to properties of the indicator and maybe network state of the animal. We conclude this because:

1. In recordings without acoustic stimulation (with the animal typically sitting quietly in the dark), the baseline varied to similar degrees as during stimulation (see Fig. R3A).
2. Across animals, the baseline during acoustic stimulation (SOA 200 Omission paradigm) sometimes increased over the course of minutes, and sometimes decreased (see Fig. R3B, different lines are different animals) with an average decrease of only about 1%.

Calcium indicators can undergo bleaching (irreversible) or dark-state transitions (reversible) which reduce their brightness. We did not observe a systematic decrease in brightness over longer periods, i.e. recordings on subsequent days showed similar absolute brightness (since there are other processes happening in parallel, i.e. continued expression as well as potential regrowth, this statement only holds from ~2 weeks after transfection to ~1 week before sacrifice, which covers the period of the presently discussed recordings). Therefore we conclude that the larger changes in baseline within a recording over multiple minutes are likely not due to these indicator-related effects, which is also in line with the fact that we tried to work with rather low light

intensities. Instead, we think these reflect neural processes, which seem to, however, not be
strongly related to the acoustic paradigm (again, see Fig. R3B), i.e. specifically they do not seem
to reflect some basic adaptation.

If we consider the animal that showed the strongest reduction in baseline fluorescence (Fig. R3C)
and then compare the ORR (red) with the PR (blue), ORR tends to be less reduced (Fig. R3C),
however, shows a bigger omission response (Fig. R3D), which again speaks against the
alternative hypothesis of a return to the original baseline at 0.
After dynamic baseline estimation (Fig. R3E) we obtain the relation in the ORR compared with
the PR (Fig. R3F) that we think best reflects their relationship.

The present, rather closely timed SOA of 200 ms hides and/or interrupts the continued dynamics
of the omission response from our view. Only at double omissions (see main manuscript, Fig. 5G)
and at the end of the stimulus, can we observe it for longer periods. Due to the length of the
stimulus sequence, we only run two trials, which means that we only have few repetitions to
inspect the omission response at the end of the stimulus sequence. We now provide this analysis
(Fig. R3G/H), which shows in both the single animal case (Fig. R3G, top static baseline) that the
omission response returns close to the moving baseline (Fig. R3G, bottom). Across animals
(Fig. R3H, static baseline) it becomes even clearer that the omission response exceeds the pre-
stimulus baseline, before returning to a lower value, close to the dynamic baseline. It is also very
noteworthy that the omission response shows repetitive peaks (Fig. R3H) after the stimulus
sequence, which are spaced approximately like the SOA, aligning with our observation in the
double omission case (Fig. 5G).

In summary, this demonstrates conclusively that the dynamic baselining strategy is the most
suitable choice for the present data, and that the omission response cannot be interpreted as a
mere return to the prestimulus baseline. Since we agree with the reviewer that this analysis is
absolutely crucial to the interpretation of the omission response, we have added this figure as a
supplementary figure to the manuscript.

 **Figure R3: Moving-median baselining effectively combats within-trial baseline drift while retaining trace shape**
 **characteristics.** All traces in the figure are derived from the PRs (blue)/ORRs (red) for the SOA 200 omission paradigm.
 **A** Baseline variations in silence are of the same order as those under acoustic stimulation (**C**). Data here shown with
 static baseline, i.e. F_0 was set to the average of the first 5s (before stimulus onset). Moving median baseline (8s window)
 overlaid as thick line, in corresponding, brighter color.
 **B** Across animals, the baseline variation under the omissions paradigm traces show different trends, with some traces
 increasing and some decreasing over time. Overall, only a small reduction in fluorescence is observed (black, gray
 indicates SEM).
 **C** Next, we show the effect of moving baselining on the animal that showed the strongest depression, here shown with
 static baseline (i.e. same principle as for silent recording in **A**). Note that the negative transient at the beginning happens
 before stimulus presentation and appears to be random. For a detailed view of the post-stimulus response see **G**.
 **D** Without dynamic baselining, the ORR and PR traces are entirely negative and offset from each other, but the omission
 response is only visible in the ORR, suggesting that the omission response is not caused by a global return to baseline
 F_0 during omissions, in particular as the more negative baseline in PR would predict a stronger return to baseline,
 which is not observed.

**E** Same data as in **C/D** with moving median baseline (8s window), globally detrends the traces, leaving local dynamics
intact.

**F** With dynamic baselining, the traces of both regions are centered on 0, while the local shape of the traces remains
largely unaffected (compared to **D**).

**G** In the example animal, offset of the stimuli (gray) causes transient activation particularly in the ORR, after which the
trace settles back towards peri-stimulation levels, consistent with the present moving baseline estimate (compare top
(static baseline) to bottom (moving baseline)).

**H** Across all animals and trials, this trend is confirmed, here shown in the static baseline: After the end of the last
stimulus, the traces settle back to the moving baseline, rather than returning to 0. In the ORR, the omission response
is clearly visible, showing multiple peaks which show similar spacing as the stimulus sequence suggesting some degree
of entrainment (however, see response to R3 on entrainment). Note that we only have 2 trials per animal, i.e. 20 trials
in total, for the present analysis, which leads to the poorer SNR here in comparison to the average responses within
the sequence.

**These signal dynamics directly affect the interpretation of the timing, magnitude, and**
**spatial localization of omission responses—the central focus of the manuscript. From a**
**conceptual standpoint, the interpretation of these responses as “prediction errors”**
**assumes that the neural signals are temporally precise and reflect a violation of an internal**
**model. However, without behaviorally relevant prediction, and in the absence of a task or**
**outcome, it is difficult to rule out alternative mechanisms such as entrainment-related**
**rebound or passive stimulus filtering, which could produce similar dynamics in the**
**absence of cognitive prediction.**

**R:** Before providing a detailed response to this point, we would like to point out that there exists
a substantial body of research on oddball responses, from multiple labs in particular Eli Nelken
and Manolo Malmierca, who work primarily in *anesthetized animals* and have convincingly linked
the increased neural activity in response to oddball stimuli to predictive principles (e.g. by the use
of control sequences etc). In the domain of omission responses, we have recently demonstrated
in a joint study with the Malmierca lab, that the omission responses are *substantially greater in*
*the awake* than in the anesthetized state (Lao-Rodriguez et al. Sci. Advances, 2023). In
accordance with this previous ePhys study, we confirm here that the omission response has a
temporally precise onset.

We would also like to reemphasize (in addition to the section in the Discussion), that the
omission response we observe is most likely consistent with an integrated mismatch signal, given
its timed onset, however, without a timed offset. An instantaneous prediction error or prediction
should decay again directly after the end of the expected, but omitted stimulus. Therefore, we see
our data and conclusions as challenging for established models of predictive processing, and
hope that they will inspire new or modified theories of the neural representation of predictions.

We agree that it would be even more convincing if a direct behavioral relation could be
demonstrated, but this is very difficult to achieve with mice in the context of fast stimulus
sequences - such as the oddball sequences - where omissions occur frequently, and the animal
would have to respond many times during the sequence. Future, modified paradigms could
resolve this problem, e.g. by creating different predictions, or spacing oddballs much further apart.

However, our case for the responses being directly related to omissions is already quite
strong, in particular because we included a broad range of controls here (in particular the double
omission and the double SOA controls), in addition to the task-free behavior analysis of the pupil
and facial movements, which clearly demonstrates tracking of recent sequence history. As we
argue in detail in the discussion, the temporal dynamics of the omission response are starkly

different from both offset responses and entrained activity (see also new figure in response to
R3), in particular showing a different, more extended progression, and we therefore see no basis
for simply equating them with either of these two. However, we agree that in particular the offset
response might be a simple variant of the omission response, which encodes the 'surprise' of a
rapid change in stimulus intensity. While therefore both offset and omission responses appear to
signal a violation of an expectation, we fully agree - and have expressed this in the discussion -
that they do not directly match predictions of theories like predictive coding (when suitably
extended in time) as the omission responses last longer than the omitted stimulus. We would
therefore argue that the omission response more likely represents an indicator of an integrated,
recent level of violation, which then also decays again (similar to the offset response, however,
the latter on much shorter timescales).

As the reviewer agrees, one of the main strengths of our methodological approach is the
ability to image the whole AC at high temporal resolution, which is a key advance, as it allows us
to compare response properties of different areas. This provides additional evidence in support
of the omission response, as it is regionally specific to a higher auditory area, while being largely
absent in the primary region.

We have checked and ensured that these arguments and distinctions receive substantial
consideration in the discussion.

**In this context, I also wonder whether the choice of imaging methods over**
**electrophysiology introduced tradeoffs that affect some of the conclusions. The widefield**
**and two-photon approaches clearly offer valuable spatial insights, particularly in**
**identifying omission response localization in ORR (likely TeA). However, for capturing**
**precise, rapid dynamics such as SOA dependence, ramping, and double-omission**
**responses at 200 ms intervals, the temporal limitations of calcium imaging may pose a**
**challenge. Even with GCaMP8m, the intrinsic integration and decay properties limit the**
**ability to resolve sub-200 ms timing features, especially without deconvolution.**

**I raise this not to question the value of imaging, but to ask whether the authors might**
**consider:**

**Showing representative raw fluorescence traces and/or overlaying $\Delta F/F$ with the baseline**
**F_0 to clarify the impact of preprocessing, exploring $\Delta F/F$ computed on a trial-by-trial basis,**
**or including deconvolved activity estimates, to better match the temporal structure of the**
**experimental design, and possibly discussing (or incorporating) complementary**
**electrophysiological recordings, even in a limited fashion, to validate key timing-based**
**inferences (e.g., omission latency, SOA scaling, ramping structure).**

**R:** Regarding the raw traces and baselining, please see above and Figure R3.

Regarding the timing the reviewer raises an important point that we have given a lot of
consideration. While imaging with GCaMP8m cannot fully match the speed of electrophysiology,
we would like to argue that it is in fact fast enough for the present requirements.

First, we would like to emphasize that we have previously demonstrated the existence of
omission responses using electrophysiology in the auditory cortex (Lao-Rodriguez et al. Sci
Advances 2023). This has also been confirmed by a second electrophysiology study recently
(Yaron et al. PLOS Biology, 2025) under similar stimulus conditions. These studies both showed
very similar onset times and time-courses, as what we presently observe, which gives us

confidence that the present GCaMP8m data is a faithful representation of the same phenomenon.
Further, a study from the lab of Eli Nelken has also provided evidence for omission responses,
although in a slightly different paradigm (Awwad et al. Current Biology, 2023). The primary
purpose of the present study was therefore the localization of these responses both in terms of
areas and layers of the auditory cortex, however, we have added a more detailed discussion of
the comparison of temporal properties between the methods to the discussion.

Second, our main subject of study are the omission responses, which we find to be
similarly slow as reported in the two studies mentioned above. Hence, the speed of the indicator
is less critical in this respect, given how fast it is (see next point). Importantly, this also means that
the area differences in relation to the omission response will remain unaffected. In terms of
resolving the stimulus driven responses, we think the widefield recordings convincingly
demonstrate that 100 Hz is sufficient to resolve the response dynamics.

Third, GCaMP8m is quite fast, maybe faster than apparent on first glance, which was
beautifully characterized in the original publication (Zhang et al. Nature Methods, 2023): The 50%
rise time of the indicator itself is ~ 2.5 ms (see Extended Data Table 6), and the 50% decay time is
~ 70 ms (assessed with a kernel fit for data from the visual cortex). Depending on the system
(culture, cortex, etc) the observed dynamics for individual neurons will differ, due to different
intrinsic calcium dynamics, and are thus slower than the indicators actual intrinsic dynamics (see
Extended Data Table 3) which has a decay of only 38 ms. Inspecting our recordings, the time
course appears to be on the fast side, closer to 50 ms for 50% decay. Together with the brightness
of GCaMP8m this in principle allows for even single spike deconvolution under optimal recording
conditions.

The main reasons why we have nonetheless not performed spike deconvolution in 2p are that:

- • Temporal resolution in 2p: assuming a decay kernel of 40-50ms, would mean that we only
have ~ 2 -3 frames per spike to deconvolve, given our 2p frame rate of 30fps, before the
signal has mostly decayed. In the study that characterized GCaMP8m, the frame rate
could be increased by focussing on a small number of cells, while here we performed
whole FOV recordings at 30fps.
- • Dynamics are qualitatively consistent: The degree of improvement that can be expected
by deconvolution depends on the kernel properties, and diminishes with faster kernels.
From the indicator properties cited above and our own observations, a best guess for the
GCaMP8m kernel would be an alpha-function with a rapid rise ($\sim t_{\text{rise}}=2.5$ ms) and rapid
decay ($\sim t_{\text{decay}}=50$ ms), overall all positive. Deconvolving with this signal, one would obtain
an estimated firing rate that will show rapid increases (positive inflections) at the identical
locations as the original signal (within the limits of our resolution). The estimated firing rate
will be more brief, although this effect will be small given our imaging rate, i.e. likely only
extend over 2-3 frames. However, no changes in polarity would be expected. In particular
for slower signals like the omission response, the difference would be minor. We support
this point, by providing deconvolved activity for a set of representative neurons (Fig. R4).
Therefore we think that the advantage offered by deconvolution would be minor in the
present setting and therefore choose to present the data in a less processed form, while
we of course fully agree to the value of deconvolution for slower indicators.
- • Avoiding assumptions: As mentioned above, the response kernel appears to differ by
system in which the recordings are performed and are a combination of indicator and

activity dynamics. By choosing a kernel to remove the indicator's dynamics, one makes
an assumption that might also lead to distortions of the data, as it might differ by region,
cell-type, etc. Avoiding such an assumption can therefore be considered a safe approach
(if the dynamics are fast enough, see next two points), which was actually strongly
encouraged by Reviewer 3.

- • Comparability of widefield and 2p data: if we were to deconvolve the data from 2p data
only, their time-courses would be harder to relate to each other. Importantly, deconvolution
should to our knowledge only be done on single cell data, as otherwise reductions in
response rate are not correctly handled, in particular after baselining in widefield.

However, in response to another question of this reviewer (see above) we have revisited our
alignment approach, which leads to an updated Fig. 2, which might already address some of the
concerns that led this reviewer to wonder about the timing. We have clarified the reasoning why
not to use deconvolution in the Methods now under the 2p data analysis.

**Figure R4: Deconvolution example** - Before averaging, raw extracted cell traces were deconvolved using MATLAB's
built-in *deconv* function with a kernel modeled as an instantaneous rise followed by an exponential decay with a time
constant of 40 ms. Example traces shown here are averaged over omissions for a single stimulus responsive cell in
PR and ORR, respectively. The deconvolution marginally sharpens the SRs, and does not abolish the OR while possibly
amplifying noise.

Omission vs. Offset Controls

**In Figure 5, the authors attempt to distinguish omission responses from offset responses
by comparing activity in a 200 ms SOA sequence (with omissions) to that following
sequence termination in a 400 ms SOA stream. However, these conditions differ not only
in the presence or absence of a final tone, but also in their temporal context—specifically,
the rate of preceding stimulation, the level of neural adaptation, and the strength of
temporal entrainment. These differences could confound any conclusions drawn from the
comparison.**

**R:** We are aware of these caveats, however, we would like to draw attention to the fact that our
study is one of few that presents multiple controls, and that each control should not be seen in
isolation, but rather in the context of all the controls. The difference in rate is indeed important to
consider, however, generally we would like to uphold that this control is quite elegant as the same,
temporally local acoustic conditions can be compared with and without expectation of a stimulus.
While had also wondered whether the difference in the rates would be a cause for concern,
however, we concluded that it remains a trustable control, as the size of the stimulus driven

response is quite comparable (see Fig. 5B), and - as we demonstrate in Fig. R3A-D - baseline
changes are not primarily driven by adaptation, and we have cases in the data set where the
average baseline of SOA200 > SOA400 and vice versa.

**For instance, fast rhythmic input (as in the 200 ms condition) can induce phase-locked**
**neural activity that gives rise to so-called “echo responses”**
**(<https://doi.org/10.1038/s41598-017-18465-w> ;) — rebound-like transients that occur at**
**expected times even after stimulus cessation, potentially reflecting rhythmic resonance**
**rather than prediction error (<https://doi.org/10.1523/JNEUROSCI.1642-19.2019>).**

**R:** We think this issue is very important, but as R2 and R3 indicate themselves, very hard to
address, also because a lot of the seeming differences are due to terminology. Specifically, echo
responses appear to be similar to the presently reported omission responses, i.e. they are quite
salient and have a specific temporal onset aligned with the expected time of the next stimulus.
However, echo responses (only?) appear for slow stimulus sequences, where we didn't observe
a substantial omission response anymore. The second citation that the reviewer provided (Latimer
et al. 2019) is very interesting and had so far escaped our attention. Their omission responses
(Fig. 7B) are indeed very reminiscent of the present data, although it lacks any kind of spatial
localization in the auditory cortex (beyond the label A1). While their approach is to fit a multi-filter
model to it (with filter lengths of ~2s), they also conclude that: "*These observations indicate that*
*adaptive processes carry information about changes in the temporal statistics of temporal*
*sequences, that is, the termination of the stimulus.*" (and they also extend this to omissions in the
subsequent section). We think one of the key concepts here is the focus on adaptive processes
that incorporate information from past experience/stimulation, which matches precisely with our
interpretation of the data. The next step in the research will be to investigate the location/sources
of these adaptive processes. As was recently shown by Tsukano et al. biorxiv (2024), even
seemingly simple adaptive processes are primarily non-local: they clearly demonstrate a top-
down circuit from the orbitofrontal cortex that targets SST cells in the AC. While we cannot draw
the same conclusion here yet, we provide a specific spatial location in secondary areas of AC,
which provides an important step towards resolving the circuit by specifically modulating inputs to
the m/pTeA or the m/pTeA itself.

We have now substantially rewrote the discussion section to point out similarities and
differences to previous studies that have shown related responses, including the references
provided by the reviewer.

**Additionally, the interstimulus response shown in the 400 ms SOA condition (Fig. 5B),**
**which the authors use as an offset control, appears to persist well into the time window**
**where an omission response would occur in the 200 ms condition. This temporal overlap**
**suggests that what is interpreted as an omission response may in fact ride on ongoing**
**offset-related activity, particularly since the omission window occurs just 200 ms after the**
**prior stimulus. This possibility deserves further attention, as it directly bears on the**
**specificity of the omission response as a prediction error signal.**

**R:** We agree that the ISR 400 activity appears not completely flat in the time period that
corresponds to the omission in SOA 200. However, the suggestion that this is "riding on the offset
response", seems very unlikely to us (see below what we precisely mean by this), due to the

substantial differences in timescale between the two. We had already noted this in the manuscript,
but had not provided a figure directly comparing them. We provide this figure now below (Fig. R5),
and will include it as a supplementary figure into the manuscript (Fig. S10).

Comparing the offset response to a single short sound (blue, 100 ms, average over 40
trials, separated by multiple seconds), to the within-sequence, after stimulus responses to the
repeated short sounds (75 ms, maroon: SOA 200, orange: ISR 400, all aligned to the sound
offset), indicates that there is fast upslope with a first peak in all traces (~70ms after stim offset),
which matches the single sound offset response dynamics. These dynamics decay after this peak
in the single sound response. However, only in the case of the SOA 200, do we see a clear,
positive inflection with at the expected time of the omission, followed by a substantial increase. In
the case of ISR400, the activity merely returns to the moving baseline (see Fig. R3 above for why
we think that baseline is the best estimate). Note, that we noticed that the data in the original
figure had used an old data export, and therefore the SOA200/ISR400 averages were slightly
different, but this has now been corrected in Fig. 5 as well. We have also verified that the post-
stimulus reduction in the case of the single sound also becomes more negative upon repeated
presentation of the stimulus (see also Fig. R1), approaching the average shape of the
SOA200/ISR400 in this first period, however, does not show an omission response (which is
expected, as here, the sound is presented in isolation with random, multisecond brakes in
between).

In summary, we conclude from the data that the offset response remains partially
identifiable even in the fast sequences, and that this argues against a generally changed shape
of it. The omission response starts later and increases for a much longer period than the offset
response, which cannot be explained by any additive effect of the offset response. We hope that
this direct comparison clarified our reasoning. We have included this figure as a new
supplementary figure (Fig. S10) and refer to it in the same place where we had previously
contrasted the shapes of the omission and offset response in the results.

**Figure R5: Comparison of offset with omission responses.** Traces were temporally aligned for
a single, 100 ms sound (light blue, same Paradigm as Fig. 1C, 40 trial average) and the OR200 (maroon, 200 trials =
omissions) / ISR400 (orange, 1440 trials = sounds) responses in the ORR of the same example animal. Within-

sequence traces were baselined using 8s moving median over the entire sequence, the Noise burst trace was baselined
to the pre-stimulus period (400ms window). Stimuli are indicated as horizontal bars, and between the vertical dashed
lines, there was no stimulus present in all conditions. Isolated sounds are followed by a fast offset response, whose
dynamics can be seen initially also in the sequence conditions. During the offset response's decay period, the within-
sequence data shows no decrease, and the OM200 shows the OR's characteristic ramping, inflecting at omission onset
(dotted vertical line), which is absent in the ISR400.

**More broadly, this raises a conceptual concern: how do we differentiate between a**
**predictive signal that reflects expectation violation, versus a biological rhythm echo or**
**offset-like signal that appears due to entrainment or rebound following regular input?**

**R:** We agree that these different interpretations are difficult to differentiate, in particular because
prediction, prediction errors, entrainment and even offset responses, while rather objectively
described, all lend themselves to interpretations in terms of predictive processing (e.g. Offset
responses can be interpreted as a deviation from a basic continuity of the sensory input), which
we also tried to emphasize in the discussion. We would therefore also advocate to take our data
at face-value, i.e. in particular the spatial localization, time-scale of response, and dependence
on a number of parameters, as shown in the controls. Future studies that modulate top-down or
local inputs will shed light on the question of the network origin and consequently their
interpretation. However, we think that the specific suggestion of entrainment can in fact be
addressed by computing measures of entrainment on the stimulus response, as suggested by
R3, which we have now performed and present in Figure S9. To avoid repeating the
data/argument here, we would kindly point the reviewer to this section of the response letter (R3,
Major point 4).

**The double-omission control helps to address some of these concerns, but its**
**interpretation may be complicated by the same $\Delta F/F$ signal dynamics noted earlier —**
**especially the strong post-stimulus suppression and potentially inflated baselines that**
**arise in high-frequency stimulus contexts. These dynamics may produce rebound-like**
**features that resemble offset or echo responses, further challenging the distinction**
**between prediction-driven and passive neural activity. Clarifying how baseline and decay**
**dynamics were handled in these controls, and providing raw or minimally processed**
**traces where feasible, would greatly aid in interpreting the origins of these signals.**

**R:** Thanks for this suggestion. We have provided these data in Figure R3 above, including details
on baseline estimation and its influence on the signals, which apply verbatim to the double
omission case. We think these address the points raised in this question.

**Jittered SOA Condition and Prediction**

**The jitter control in Figure 6 is conceptually valuable and well-designed in intent, but I**
**found it difficult to interpret the presence of any omission response under this condition.**
**The jittered sequence uses a pseudorandom set of SOAs (100–300 ms), eliminating**
**consistent temporal regularity. From a predictive coding perspective, one might not expect**
**strong omission responses in the absence of a stable rhythm to generate precise temporal**
**predictions.**

**The authors do report reduced ORs under jitter (Fig. 6B), but not abolished. The presence**
**of even reduced omission responses in this context raises a question: are these signals**

**truly reflecting a top-down temporal prediction error, or do they reflect bottom-up**
**adaptation or rebound dynamics that are less sensitive to fine-grained regularity? Without**
**a behavioral marker of prediction or evidence of hierarchical feedback, it is difficult to**
**conclude that these signals represent cognitive prediction violations rather than statistical**
**learning or passive entrainment. Additionally, the same $\Delta F/F$ signal dynamics described**
**above (e.g., suppression, baselining issues) may behave differently under regular vs.**
**jittered timing, potentially exaggerating this apparent drop.**

**If the omission response reflects a rebound or offset-like phenomenon, this could**
**confound the intended interpretation of the jitter control as evidence for prediction-**
**specific processing.**

**R:** We were also initially surprised by the fact that the jitter control only led to a reduction, but no
abolishment of the omission response. In hindsight, it, however, can be understood in the
framework of predictions, as there remains a prediction that a sound of the same frequency will
appear within an approximate, limited time-window, which is much more concrete than no
prediction. However, we agree that this is not the only possible interpretation. We agree that
entrainment could be considered as an alternative, however, the time-course of the omission
response and the stimulus-driven response are so different in size and dynamics, that classical
entrainment appears unlikely, which is further detailed in the entrainment analysis we performed
(Fig. S9). We do not comment here on baselining/suppression and refer to the response given
above, where in particular we have argued that the local response shape is unaffected by
baselining and that the omission response 'returns' to the moving baseline, not the prestimulus
baseline. About statistical learning: we are not sure whether that would be incompatible with
prediction-processing. In fact, we would argue that statistical learning implies that the system
adapts to some part of the stimulus statistics and stops responding to it, which immediately implies
that responses are then related to aspects of the sensory environment that have not been learned,
i.e. are unpredicted. We have tried to clarify this further in the discussion, also in particular that
we are not tied to any particular theory of predictive processing or statistical learning.

**Statistical Framework**

**A few statistical issues are worth revisiting:**

**MANOVA (Fig. 2H) assumes multivariate normality and independence, which is difficult to**
**justify in calcium imaging. Non-parametric alternatives may be more robust here.**

**R:** We had originally checked the degree of normality/log-normality of the data, for which we used
the Shapiro–Wilk test combined with the Quantile-Quantile plot (without or with log-transforming).
In 16 out of 24 distributions it did not significantly differ from normality and we therefore assumed
that the distributions are typically normally distributed. Since we could not locate a suitable test in
Matlab to perform the non-parametric equivalent of the hierarchically structured test (see next
question), we kept the normality assumption, in particular given the tiny p-values ($\sim 10^{-10}$) the
specific assumption of the test would very likely give the same result.

**The data are hierarchically structured (neurons within animals), yet most tests treat each**
**neuron as independent. Animal-level averaging or mixed-effects models would more**
**accurately reflect the design and reduce pseudo-replication.**

**R:** Thanks for pointing this out. We indeed ran all tests on the combined set of neurons, but agree
that a mixed effects model is more appropriate in this analysis. We use a linear mixed effects
model now, which only led to changes in the specific p-values, but no qualitative changes in the
results (as expected). The Methods section has been adapted accordingly, as well as the
corresponding figures and Results sections.

**Behavioral Measures (Figure 4)**

**The inclusion of facial motion and pupil responses is an important strength, but I would**
**encourage a more cautious interpretation:**

**Facial motion increases after omissions but is time-locked to the next sound, not the**
**omission itself. This makes it difficult to argue that the movement reflects prediction error**
**rather than rebound excitation or adaptation. There is also no reported correlation between**
**motion energy and neural responses.**

**R:** Indeed, we agree and we also did not claim that facial motion is reflecting a prediction error to
the omission. Instead we argue that the adaptation of the size of the facial motion as a function
of the number of sounds (Fig. 4F) and its recovery in size after an omission (Fig. 4E), show that
facial motions are not a hardwired reflex to a sound, but show a dependence on the recent
sequence statistics. Given the pattern of response to individual sounds, i.e. rapid after about
70 ms (matching Clayton et al. 2024) and a transient response (see also Fig. 4B bottom), and the
lack of a response to the omission (within the time window expected from the stimulus response),
we think that the hypothesis of a rebound excitation is unlikely.

Regarding the relation between motion energy and neural response: We performed two
analyses to address this question.

First, we find that the size of the neural response shows the same pattern after the
omission, i.e. first one much larger, and then adapting to a lower value (see Figure R6 below, left
panel). Overall this mirrors the dependence we have reported before using electrophysiology
(Lao-Rodriguez et al. 2023), however, here we in addition report that the adaptation of the
response is stronger in the omission response area (see below and new panel Fig. 4H). We
document this now in the associated Results.

Second, we find that the time-series of the two measures correlate significantly as well
(see Figure R6 below, right panel), although substantial variability exists here. Interestingly, again
the ORR shows a higher degree of correlation. We now mention these correlations in the text.

**Figure R6: Relation between Facial Motion and Neural Responses to Standard Stimuli**

**A - Post-omission progression of neural standard responses.**

The neural response size to the standard sounds (SR) decays similarly, although slightly faster than the
 facial motion (see Fig. 4F). The decay is stronger in the omission responsive region (ORR) than in the
 primary region (PR).

**B - Correlation between standard response in facial motion and neural data.**

SR magnitude for primary region & Face are positively correlated (Pearson correlation 0.116 and 0.229 for
 PR and ORR, respectively) across single trials, for stimuli up to 4 standards after omission.

**Pupil dilation peaks ~500 ms after the omitted stimulus—well after the neural omission**
 **response. This is compatible with a general arousal response, possibly LC-NE-mediated,**
 **but not direct evidence for predictive monitoring. Moreover, the dilation overlaps with**
 **responses to subsequent stimuli, further complicating interpretation. These signals may**
 **be better framed as nonspecific arousal correlates rather than behavioral readouts of**
 **predictive timing, particularly given the lack of task engagement.**

**R:** We agree that the timing of the signal *peak* is delayed, however, the *time of onset* is quite
 clear, and the time-course of pupil responses is known to be comparatively slow, as we and others
 have shown before (half-max rise time ~0.5s, i.e. very similar in time-scale to the present results,
 e.g. Fig. 1E (Clayton et al. Current Biology, 2024) or Fig. 3C bottom (Alishbayli et al., 2025). To
 support this further quantitatively we have statistically compared the size of the pupil after the
 omission, but before the next stimulus (110 ms after omission onset) to the pupil size 50 ms before
 the omission, which already reaches significance ($p=0.027$, Wilcoxon test). While we think this
 provides solid evidence that pupil dilation is indeed triggered by the omission, rather than the
 subsequent sound (in which case we would have expected an onset at least 200 ms post-
 omission). We would like to emphasize that our overall qualification of these behavioral results is
 already quite cautious, i.e. that there are behavioral responses that indicate a certain degree of
 monitoring of the sequence statistics. However, we agree that only an experiment involving active
 behavior can provide more clarity here. The challenge that we see in setting up this experiment
 is that the animal would have to respond multiple times in the sequence, and response times

would have to be rapid to reliably link it to the omission, rather than the next sound. Another
possibility would be to change the stimulus paradigm, e.g. to the global-local paradigm, and
randomly omit the last stimulus in a shorter sequence of stimuli. To address the reviewers
comment we provide the above reasoning and additional test in the Results and have emphasized
the need for additional behavioral confirmation in the Discussion.

**Reviewer #3 (Remarks to the Author):**

**This manuscript by Peters, Cai et al. describes results of experiments designed to**
**investigate the neural representation of sound omissions (unexpected absences from an**
**otherwise statistically predictable sound sequence) in the auditory cortex of awake mice.**
**Neural responses to unexpected omissions are of great interest to sensory neuroscientists**
**because they arguably represent the purest form of evidence for statistical learning and/or**
**sensory prediction; there is no stimulus present during an unexpected omission and yet**
**the omission evokes neural activity. Omission responses have been widely demonstrated**
**in humans using EEG and also, in a few studies, in animals including rats and mice.**
**However, the neural mechanisms underlying omission responses and their localisation**
**within the brain have remained largely open questions. The answers to these open**
**questions have profound implications for prominent theories about "predictive coding" in**
**the brain, with potential to help resolve debates about the extent to which neural responses**
**in different cortical and subcortical regions represent stimulus features, sensory**
**predictions, prediction errors, or something else.**

**Here, experiments were conducted in awake passively listening mice, while neural activity**
**across primary and secondary fields of auditory cortex was monitored using widefield**
**imaging and, at the single-neuron level in more targeted fields of view, using two-photon**
**imaging with an fast calcium indicator, jRCaMP1e. Facial movements were measured**
**during the experiments, providing a spontaneous behavioural measure of arousal. Multiple**
**stimulus paradigms were used to rule out alternative explanations for the results and to**
**test predictions of prominent theories about neural representation, such as that omission**
**responses signal sensory predictions or prediction errors.**

**The manuscript makes at least three major contributions to systems neuroscience:**

**(A) Results clearly demonstrate the existence of a secondary region of the auditory cortex**
**in mice (pTeA) in which neurons with strong omission responses are clustered. There are**
**multiple well-designed controls that effectively rule out alternative explanations, and some**
**of these controls are quite novel and innovative in their own right (e.g. the second-**
**omission control, which could also be useful for investigating omission responses in**
**human EEG studies).**

**(B) Intriguingly, the temporal characteristics of the omission responses observed in the**
**mouse brain are inconsistent with some theories of predictive coding. In particular, the**
**observed responses ramp up monotonically following the unexpected omission rather**
**than following the expected timecourse of the omitted stimulus, implying that the**
**responses represent something more like an *integrated* prediction or prediction error**
**signal. This result provides an important new constraint for theories of sensory**
**representation in the auditory cortex, and for analysis of "mismatch responses" and other**
**deviant responses in all sensory modalities.**

**(C) Finally, the manuscript serves as an excellent early illustration of the power of fast two-**
**photon imaging techniques to address questions about neural representation on ~10ms**
**timescales. The authors used a relatively new and very fast calcium indicator (jRCaMP8m)**
**and rigorously minimised the use of deconvolution and filtering, and thereby provide a**
**very robust demonstration that calcium imaging can be used effectively to explore**
**questions about temporal representation of sound signals that would previously have**
**been robustly accessible only using electrophysiological techniques.**

**The manuscript is also very well-written. The authors should be particularly commended**
**on the thoughtfulness of their Discussion section, which carefully lays out the evidence**
**against possible alternative explanations for the key observations, and explores a wide**
**range of implications of the results for systems, cognitive and computational**
**neuroscience.**

**R: We would like to thank the reviewer for the supportive perspective on our study!**

**MAJOR COMMENTS:**

**1. Data on the laminar distribution of omission responses vs standard responses is less**
**convincingly presented than the other results. In particular, Figure 3 is not fully explained.**
**What do the coloured vertical strips in the top lineplots of Figure 3A represent? What are**
**the axes of the matrix plots in Figure 3A? More critically, why is the "omission response"**
**line in the Figure 3C not normalised to 100% at 100um, given that the x-axis label says**
**"Response rel. OR at 100um"? While the other figures in the manuscript were generally**
**very convincing and largely self-explanatory, Figure 3 is not convincing in its present form.**

**R: Thanks for pointing this out. We have substantially improved Figure 3, by adding a new**
**introductory panel that shows the depth stack that we performed. Further, we now closely match**
**the style of the schematics in Fig. 3B to the schematics in other figures, using a consistent color**
**scheme throughout for omission and standard response related plots as well as connecting**
**elements in the plot better, in particular to the sample data plots below. The normalization was**
**checked, it is done with respect to OR in ORR at 100 μ m (adapted in the figure). Further, we have**
**adjusted the labels to be more clear and a more detailed explanation of the depth dependence**
**was provided in the text.**

**2. Several controls provide compelling evidence demonstrating that activity in the ORR is**
**related to omissions rather than offset responses (e.g., comparison between omissions at**
**200ms SOA and expected sounds at 400ms SOA, and observations regarding the double-**
**omission response). However, the claim that the ORR can be clearly distinguished**
**topographically from regions with offset responses could be better supported. Please**
**provide more information about exactly how offset response magnitude in Supplementary**
**Figure 3 is defined. Peak $\Delta F/F$ over what time interval after offset? Robust identification**
**of offset responses is not as easy as for onset responses because an excitatory offset**
**burst can be difficult to distinguish from a slow decay in sustained activity. There's a lot**
**of overlap in the onset and offset response maps in Supplementary Figure 3; this is**

potentially unsurprising as a lot of auditory cortex neurons have responses to both
transients. However, another possibility is that the offset response measure is slightly
confounded here with the onset response measure, because a cell with high sustained
response that decays slowly following sound offset might show up as having both a large
onset response AND a large offset response, even in the absence of a "true" offset
response burst. What would maps of the difference in $\Delta F/F$ between the offset period and
onset (or onset+sustained) period look like? More generally, how do the authors know that
their offset response measure is capturing "true" offset responses rather than slow decays
of sustained activity?

**R:** The reviewer is emphasizing an important question here, which has also been at the center of
our considerations. We would like to reemphasize that in addition to the differences in spatial
location, the time-scales of offset responses and omission responses are very different, with the
former peaking around 50 ms and then decaying (Fig. 1C, and see (Olsen & Hasenstaub, 2022),
Fig. 2E_{vii} for single cell data), while the latter continue to increase at least up to 400 ms and does
not show a clear peak up to this point. Furthermore, the OR shows a dependence on time-history
(shown now in Fig. 4H), which is also uncharacteristic of "simple" bottom-up offset responses
(Olsen & Hasenstaub, 2022). However, as pointed out by the reviewer, slow response dynamics
and activity summation can be important confounds when evaluating offset responses, especially
for the short (75 ms) stimuli used here. To reduce this effect, we have adapted Fig. S3 to show
the Offset response relative to the last ΔF value before stimulus offset, rather than the pre-stimulus
ΔF value. The resulting map still clearly differs from the OR map, highlighting AAF and an
uncharacterized antero-medial area, but there is also overlap of the two responses in pTeA/mTeA.

In summary, and in accordance with our detailed assessment of this question in the
discussion, we conclude that offset responses could be a very basic, temporally local version of
the presently observed omission responses, but that the temporal characteristics are so different
that substantial, context-dependent modification is necessary to get from one to the other. Hence
both of them could still represent a violation of an expectation - i.e. offset responses for the
immediate violation in intensity, and omission responses for a delayed, temporal expectation -
however, different experiments that modulate top-down inputs (e.g. the excellent study by
Tsukano et al. bioRxiv, 2024) will be necessary to more clearly separate these. We have adapted
Fig. S3 accordingly, as well as the Methods and the associated description in the Results and
Discussion.

**3. Also regarding maps in Supplementary Figure 3 (and elsewhere): There seems to be a**
**very prominent outlining of blood vessels in the response latency, response modulation,**
**onset response, offset response and omission response heat maps. Why might this be?**

**R:** Thanks for pointing this out, which had escaped our attention. Investigating the source of the
vessel outlines revealed that these were driven by small motion artefacts (typically 1 pixel or less),
which caused the ΔF -derived metrics on vessel-bordering pixels to be strongly influenced. To
calculate these response maps, we now implemented sub-pixel rigid motion correction
(NoRMCorrE package, Pnevmatikakis & Giovannucci, 2017), which strongly reduces these
motion artefacts. Fig. S3 and Fig. 6C, as well as the Methods section have been updated
accordingly.

**4. The dependence of omission responses on SOA was non-monotonic with SOA (Figure**
**5O). Also, the dependence of omission responses on the number of previous**
**presentations of regular stimuli was slightly non-monotonic (increasing to 40-70 trials then**
**slightly decreasing). Both of these observations would seem to suggest that**
**entrainment/oscillation phenomena might be a more accurate and parsimonious**
**explanation for the observed omission responses than integrated prediction or prediction**
**error. Granted, these explanations are not possible to disentangle cleanly, but the non-**
**monotonicities provide an intriguing avenue for further investigation. Do measures of**
**entrainment in the responses to non-omitted stimuli show the same profile as a function**
**of SOA or number of previous stimuli as the omission responses do? If so, this might be**
**good evidence for the entrainment hypothesis --- or at least, good evidence for circuit-level**
**constraints on the more cognitive predictive-coding interpretations, which would not**
**predict non-monotonicity with SOA or number of previous stimuli.**

**R:** This comment was really inspiring to us, as we had previously deemed signatures of
entrainment hard to disentangle from those of predictive processes. However, upon further
consideration and analysis we think we can provide an answer to this question. The reviewer is
specifically asking whether the observed omission response could be a consequence of
entrainment. We argue below that this is not supported by our data, however, that there appears
to be entrainment of the stimulus driven response as a function of time, after an initial adaptation.

To arrive at these conclusions, we explored multiple measures of entrainment (coherence,
peak-to-trough response size, phase-lag). Upon inspection of the data (see figure R7 below), we
concluded that peak-to-trough size of the response within a limited time window matched to the
peak and trough of the average response (which is dominated by the majority of the later
responses) gave the most reliable estimate of how the response size changed as a function of
continued exposure to the stimulus (Measures assuming a sinusoidal response shape were
avoided, as the response was not clearly sinusoidal and there was no reason to assume a basic,
sinusoidal shape). This also allowed us to compare the omission response's size in relation to the
average, possibly entrained shape of the stimulus response. Further, we estimated a top and
bottom envelope through these peaks and troughs of the stimulus response (Fig. R7A, orange
and yellow), whose average was used as an estimate of the overall response size/adaptation over
time.

As a function of time into the trial (i.e. number of presentations of the standard stimulus),
we observe two patterns:

- ● The overall response size rapidly reduces over the course of ~10 stimuli (Fig. R7C)
- ● The peak-to-trough size increases with a slightly slower time-course (Fig. R7F) and then
stays at this level (Fig. R7D/E)

Further, we find that:

- ● The omission response itself does not change its shape to match the stimulus response
(summary in Fig. R7H, and all main figures showing omission responses) and therefore
its values are very low in comparison.
- ● After an omission, the entrainment of the stimulus responses is not reduced but is even
larger for the next stimulus, i.e. very different from the slow build up at the beginning of
the stimulus (Fig. R7G).

- ● Across SOAs, the shape of the degree of stimulus entrainment of the omission responses
is similar to what we observed for the size of the omission response, however, this is also
somewhat a tautology given that the entrainment measure is also measuring the size of
the omission response, however, at select time points.

In summary, using the peak–trough difference as our entrainment measure, we found that
stimulus responses, particularly in the ORR, exhibited a clear entrainment build-up over the first
$\sim 15 \pm 5$ stimuli of the sequence. However, applying the same measure to the omission response
quantifies the fact that the shape of it is very different from the stimulus response, and no
convergence of the omission responses to the entrained stimulus response is observed, which
excludes it as an entrained aftereffect of the stimulus response in our assessment.

We have now added this as a supplementary figure in the manuscript (Fig. S9) including
a corresponding description in the Results section and discuss the possibility in detail in the
Discussion in the section on alternative explanations. We would be happy to add it as a main
figure, but are unsure whether this would be allowed by the word count restrictions.

**Figure R7: Sound responses, but not omissions, are subject to entrainment across areas.**

Entrainment score for each stimulus was defined as the amplitude difference between the response peak
and trough within an empirically selected window starting at stimulus onset and going to 50 ms after stimulus
offset (for 100 ms SOA, only 25 ms), measuring the similarity to the average stimulus response shape, after
normalization (see below).

**A & B** Example single-animal, raw $\Delta F/F$ traces aligned to sequence onset for PR (A) and ORR (B)
respectively. Envelopes were drawn with interpolation between peaks (orange) and troughs (yellow) as a
visual indicator of entrainment score.

**C** The mean overall response was calculated by first averaging the upper and lower envelopes per animal,
and then computing the mean across animals. For both areas, following the initial increase, overall
response size rapidly reduces over the course of ~ 10 stimuli (2s).

**D & E** Mean, normalized entrainment score across all animals over an entire trial for PR (D) and ORR (E)
respectively. Following an early, fast rise and slower decline, scores stabilized with fluctuations around a
steady level. Normalized entrainment score for each animal was obtained by dividing its raw entrainment
score by the median raw entrainment score of the primary region across all stimuli.

**F** The initial entrainment increase at the start of the stimulus sequence. Solid lines indicate linear fits of
entrainment scores. ORR exhibited a faster and more pronounced increase than PR.

**G** Mean entrainment scores were computed for omissions and the stimuli following them, averaged across
all omission events. Entrainment scores were significantly lower during omissions periods compared with
stimulus periods. The first stimulus after an omission exhibited significantly higher entrainment than
baseline, followed by a rapid return to baseline levels.

**H** Distribution of entrainment scores during stimulus and omission periods was assessed across both areas
and all animals. In both areas, entrainment scores during stimuli were significantly higher than during
omissions ($p < 10^{-4}$, custom permutation test), whereas asymptotic entrainment values did not differ
significantly between areas.

**I** Entrainment scores were computed across paradigms with different SOAs. Scores increased significantly
from 100 to 200 ms SOA ($p < 10^{-4}$, custom permutation test), and remained stable between 200 and 400 ms
SOA. This pattern is partly consistent with Fig. 5Q.

The number of animals used in this figure is 7 (N=7).

**MINOR COMMENTS:**

**Abstract: "Animals were aware of omissions..."** Aware is something of a loaded word and
should be avoided. It is possible that the arousal responses measured in spontaneous
behaviour are entirely subcortical and only minimally relevant to "awareness".

**R:** Good point, we have reworded this in the abstract to the following sentence:

"Omissions and sequence statistics were reflected in behavioral responses, namely timed pupil
dilation and rapid facial motions, respectively."

**Abstract: I think the abstract could be slightly improved to highlight the significance of**
**this paper more clearly. In particular, one of the greatest strengths of this paper is the**
**careful comparison of different stimulus paradigms (varying SOAs, jitter, number of**
**omissions etc) to rule out offset responses or other possible explanations for the observed**
**omission responses. This is summarised mainly in the sentence about omission**
**responses being "temporally distinct" from offset responses, which doesn't really do the**
**authors' efforts here full justice.**

**R:** Thanks for pointing this out. Within the word-count limit of the abstract we have extended a
sentence to put more emphasis on this, i.e. "Omission responses were also distinct from offset
responses in temporal dynamics and size, as verified by multiple controls."

**Figure 4, panel O: This plot is described in the legend as showing a "low-pass**
**dependence" of OR on SOA, but in fact it is bandpass with peak at 200ms. Why might this**
**be? More explanation or at least speculation here would be helpful. The 200ms SOA almost**
**seems a bit of a "sweet spot". In fact for SOAs above 300ms it seems that the omission**
**response is plateauing. Could this reflect an interaction with brain oscillations, or (perhaps**
**equivalently) with short-term synaptic plasticity timescales?**

**R:** Agreed, we were likely biased in our description by the knowledge of the human studies and
the fact that in our previous electrophysiology study we had observed the strongest effect at
125ms SOA. Our choice of 200 ms SOA for the present study was chosen to account for the time-
constants of the calcium indicator, but before knowing the dependence on SOA reported here.
We have now adapted the description of the graph to indicate a bandpass characteristic.
Regarding the alternative hypotheses, we have extended our discussion on the possibly
underlying mechanisms in this direction, concluding that both brain-oscillations (here: delta (0.5-
4 Hz) and ~theta (4-8 Hz) bands) as well as short-synaptic plasticity might indeed contribute (see
the work of Markram et al. 1998, Fig. 4). We discuss this now briefly in relation to a possible role
of entrainment.

**Figure 6: It's very intriguing that the effect of sequence jitter on the omission responses**
**was so minimal once the impact of sequence jitter on baseline activity was taken into**
**account. This observation seems itself worth a bit more attention in the Discussion.**

**R:** We agree completely, and it was also our hypothesis that we would see a stronger effect here.
Our post-hoc interpretation of this observation is that while this reduces the temporal predictability
of the sound, the spectral predictability remains the same. We have added a section to the
discussion including an overview of the related human literature.

**Line 57: "tone height" -> tone frequency**

**R:** Thanks, changed.

**Line 59: "predictive processings" -> predictive processing**

**R:** Thanks, changed.

**Line 74: "Calcium" should be lowercase**

**R:** Thanks, changed throughout the manuscript.

**Discussion: Generally the discussion is excellent (especially the subsections on**
**alternative explanations for omission responses and implications for predictive coding).**
**However, the first subsection on the relation to previous studies could be improved; it**
**doesn't flow very well across some seemingly disjointed paragraphs. For example the**
**paragraph about whether omissions are regularly or randomly selected (lines 402-405)**
**feels a bit disconnected from everything else, and it's not really made clear what random**
**or regular selection of omissions means.**

**R:** Thanks for suggesting this improvement for the organization of the Discussion section. Upon
rereading this section, we agree that at least two paragraphs were not optimally embedded. We
have now added 4 headings to group the points in this discussion part into Temporal, Spatial,
Layer, and Contextual comparisons. We think this has helped to clarify this part of the discussion.

**The Discussion would also benefit from a bit more information about how the areas known**
**to connect to ORR might contribute to generation of the monotonically ramping omission**
**responses (and/or changes in baseline activity with jittering of regularity, which was also**
**a very intriguing observation in itself).**

**R:** Thanks for pointing this out, in particular as there is suggestive evidence in this direction on
the basis of a few recent preprints and publications, e.g. (Obara et al., 2023; Tsukano et al., 2024).
We have added a corresponding subsection to the Discussion.

Bibliography

**Alishbayli, A., Przewrocki, K., Heumen, P. van, & Englitz, B. (2025).** *Neural integration of*
*acoustic statistics enables detecting acoustic targets in noise* (p. 2025.05.29.656794). bioRxiv.
<https://doi.org/10.1101/2025.05.29.656794>

**Jamali, S., Bagur, S., Bremont, E., Van Kerkoerle, T., Dehaene, S., & Bathellier, B. (2024).**
*Parallel mechanisms signal a hierarchy of sequence structure violations in the auditory cortex.*
*eLife*, 13, RP102702. <https://doi.org/10.7554/eLife.102702>

**Li, J., Liao, X., Zhang, J., Wang, M., Yang, N., Zhang, J., Lv, G., Li, H., Lu, J., Ding, R., Li, X.,**
**Guang, Y., Yang, Z., Qin, H., Jin, W., Zhang, K., He, C., Jia, H., Zeng, S., ... Chen, X. (2017).**
*Primary Auditory Cortex is Required for Anticipatory Motor Response.* *Cerebral Cortex (New York,*
*N.Y.: 1991)*, 27(6), 3254–3271. <https://doi.org/10.1093/cercor/bhx079>

**Obara, K., Ebina, T., Terada, S.-I., Uka, T., Komatsu, M., Takaji, M., Watakabe, A., Kobayashi,**
**K., Masamizu, Y., Mizukami, H., Yamamori, T., Kasai, K., & Matsuzaki, M. (2023).** Change
detection in the primate auditory cortex through feedback of prediction error signals. *Nature*
*Communications*, 14(1), 6981. <https://doi.org/10.1038/s41467-023-42553-3>

**O’Connell, M. N., Falchier, A., McGinnis, T., Schroeder, C. E., & Lakatos, P. (2011).** Dual
Mechanism of Neuronal Ensemble Inhibition in Primary Auditory Cortex. *Neuron*, 69(4), 805–817.
<https://doi.org/10.1016/j.neuron.2011.01.012>

**Olsen, T., & Hasenstaub, A. R. (2022).** Offset Responses in the Auditory Cortex Show Unique
History Dependence. *The Journal of Neuroscience*, 42(39), 7370–7385.
<https://doi.org/10.1523/JNEUROSCI.0494-22.2022>

**Pnevmatikakis, E. A., & Giovannucci, A. (2017).** NoRMCorre: An online algorithm for piecewise
rigid motion correction of calcium imaging data. *Journal of Neuroscience Methods*, 291, 83–94.
<https://doi.org/10.1016/j.jneumeth.2017.07.031>

**Sadagopan, S., Kar, M., & Parida, S. (2023).** Quantitative models of auditory cortical processing.
*Hearing Research*, 429, 108697. <https://doi.org/10.1016/j.heares.2023.108697>

**Tsukano, H., Garcia, M. M., Dandu, P. R., & Kato, H. K. (2024).** *Predictive filtering of sensory*
*response via orbitofrontal top-down input.* <https://doi.org/10.1101/2024.09.17.613562>

**Voytenko, S. V., & Galazyuk, A. V. (2010).** Suppression of spontaneous firing in inferior colliculus
neurons during sound processing. *Neuroscience*, 165(4), 1490–1500.
<https://doi.org/10.1016/j.neuroscience.2009.11.070>

**Willmore, B. D. B., & King, A. J. (2022).** Adaptation in auditory processing. *Physiological Reviews.*
<https://doi.org/10.1152/physrev.00011.2022>

REVIEWERS' COMMENTS

Responses from the authors formatted in blue.

Reviewer #1 (Remarks to the Author):

The authors have fully addressed all my comments.

The authors thank the reviewer for their critical and constructive feedback.

Reviewer #2 (Remarks to the Author):

The authors have made substantial efforts addressing my concerns by providing more analyses and clarifications. The spatial findings—TeA localization, layer-specific distribution, and hierarchical organization—are robust and represent valuable contributions to understanding predictive processing. However, claims about temporal precision throughout the manuscript exceed what the methods can establish. The authors themselves acknowledge: "While it cannot be concluded with certainty..." (response line 190-191) and note "baseline level is hard to determine during continuous stimulation" (response line 216). Given 2-photon imaging at 30 Hz (33ms resolution) and jGCaMP8m decay of ~40ms, they observe temporal structure at this scale but cannot prove neural computation precision versus measurement ceiling.

I would suggest the following text changes:

Line 15-16 (Abstract): Change "Neuronal responses to the omission of expected sounds were precisely timed" to "Neuronal responses to the omission of expected sounds were time-locked to expected stimulus onset"

Done.

Lines 82-83, 130, 252 (Results): Replace all instances of "precisely timed" with "time-locked to expected onset"

Done where applicable.

Lines 408-414 (Discussion): After "temporally precise (10s of ms) mechanism," add: "While jGCaMP8m provides improved temporal resolution, the precise timing of predictive computations at the millisecond scale remains to be confirmed with electrophysiology"

Done.

Line 475 (Discussion on offset responses): Change "effectively ruled out" to "substantially reduced the likelihood" (authors acknowledge: "While not impossible, we consider this rather implausible" - response line 484)

Done.

Lines 127-128, 157-158 (OR magnitude): Add caveat that magnitude estimates depend on baseline choice during continuous stimulation

Done, thanks.

Lines 89-92 (Abstract, behavioral): Change "reflected

in behavioral responses" to "correlated with behavioral changes"

Done, thanks.

Lines 583-587 (Discussion, limitations): Add that active paradigms are needed to distinguish prediction monitoring from arousal responses (authors state: "only an experiment involving active behavior can provide more clarity" - response line 846)

Added to Discussion "[...] and enable distinction of prediction monitoring from arousal responses" to the sentence where this is semantically appropriate.

With these revisions, I recommend publication.

The authors thank the reviewer for their critical and constructive feedback.

Reviewer #3 (Remarks to the Author):

The authors have done a very thorough job of addressing my comments and I have no further suggestions for revision. Thank you for the detailed explanations and additional analysis.

The authors thank the reviewer for their critical and constructive feedback.

Regarding the authors' query about whether their new Figure S9 addressing questions about entrainment should be added as a main figure --- I think it's probably best as a supplementary figure given the level of detail, but it is a very nice addition nonetheless.

We agree, and will keep Figure S9 in the Supplement.